



# Fault sealing and caprock integrity for CO₂ storage: an in-situ injection experiment

Alba Zappone[1,2], Antonio Pio Rinaldi[1,5], Melchior Grab[3], Quinn Wenning[3], Clément Roques[3,4], Claudio Madonna[2], Anne Obermann[1], Stefano M. Bernasconi[3], Florian Soom[5], Paul Cook[5], Yves Guglielmi[5], Christophe Nussbaum[6], Domenico Giardini[3] , Stefan Wiemer[1]

[1]Swiss Seismological Service, ETHZ, Zurich, 8092, Switzerland
[2]Department of Mechanical Engineering, ETHZ, Zurich, 8092, Switzerland
[3]Department of Earth Sciences, ETHZ, Zurich, 8092, Switzerland
[4]Géosciences Rennes, University of Rennes 1, Rennes, 35000, France
[5]Energy Geosciences Division, LBNL Berkeley, CA 94720, USA
[6]Swiss Geological Survey, swisstopo, Wabern, 3084, Switzerland

*Correspondence to*: Alba Zappone (alba.zappone@sed.ethz.ch)





**Abstract.** The success of geological carbon storage depends on the assurance of a permanent confinement of the injected $CO_2$ in the storage formation at depth. One of the critical elements of the safekeeping of $CO_2$ is the sealing capacity of the caprock overlying the storage formation, despite faults and/or fractures, which may occur in it. In this work, we present an ongoing

injection experiment performed in a fault hosted in clay at the Mont Terri underground rock laboratory (NW Switzerland). The experiment aims at improving our understanding on the main physical and chemical mechanisms controlling i) the migration of $CO_2$ through a fault damage zone, ii) the interaction of the $CO_2$ with the neighbouring intact rock, and iii) the impact of the injection on the transmissivity in the fault. To this end, we inject a $CO_2$-saturated saline water in the top of a 3 m think fault in the Opalinus Clay, a clay formation that is a good analogue of common caprock for $CO_2$ storage at depth. The mobility of the

$CO_2$ within the fault is studied at decameter scale, by using a comprehensive monitoring system. Our experiment aims to the closing of the knowledge gap between laboratory and reservoir scales. Therefore, an important aspect of the experiment is the decameter scale and the prolonged duration of observations over many months. We collect observations and data from a wide range of monitoring systems, such as a seismic network, pressure temperature and electrical conductivity sensors, fiber optics, extensometers, and an in situ mass spectrometer for dissolved gas monitoring. The observations are complemented by

laboratory data on collected fluids and rock samples. Here we show the details of the experimental concept and installed instrumentation, as well as the first results of the preliminary characterization. Analysis of borehole logging allow identifying potential hydraulic transmissive structures within the fault zone. A preliminary analysis of the injection tests helped estimating the transmissivity of such structures within the fault zone, as well as the pressure required to mechanically open such features. The preliminary tests did not record any induced microseismic events. Active seismic tomography enabled a sharp imaging

the fault zone.

KEYWORDS: $CO_2$ STORAGE: $CO_2$ INJECTION, CAPROCK INTEGRITY, SITE MONITORING, IN-SITU DECAMETER SCALE EXPERIMENT

**1 Introduction**

Carbon Capture and Storage (CCS) has a fundamental role to reduce the content of anthropogenic $CO_2$ in the atmosphere and to achieve the Paris Agreement's challenging objective of keeping a global temperature rise below 2°C above pre-industrial levels (IPCC 2018, 2019; Cozier, 2015).

Carbon storage at mega-tons scale has been proven successful (e.g., Sleipner, Snøhvit, Weyburn, Aquistore, Quest), but needs

to be increased to giga-tons scale in order to achieve global emissions reductions targets (IPCC, 2018; Zoback & Gorelick,





2012). Achievement of this upscaling is critically linked to better estimates of storage capacity and improved risk management strategies that rely on detailed monitoring with combined geophysical, geochemical, and hydrogeological methods (e.g. Aagaard et al., 2018; Fang et al., 2010; Rutqvist, 2012). One of the challenges both in evaluating the storage capacity and in pressure-managing strategies is the assessment of long-term integrity of sealing formations. $CO_2$ leakage along potential high
permeability pathways into near-surface aquifers or to the surface is potentially one of the main geological hazard for CCS that might challenge the social and political acceptability of the whole technology (Zoback and Gorelick, 2012).

Faults within the caprock represent one of the possible pathways for $CO_2$ to migrate out of the storage reservoir. The presence of faults will greatly affect the site characterization process in terms of safety assessment, and consequently in the monitoring plan, and risk management (prevention, mitigation, remediation actions). Faults are also key elements in the evaluation of
induced seismicity risk during injection operations (Rutqvist et al., 2016). It was argued that the injection of large volumes of $CO_2$ at relatively shallow depth (few km) in brittle rocks could trigger earthquakes (Zoback and Gorelick, 2012), although injecting in soft sedimentary basin (Vilarrasa and Carrera, 2015) might reduce such potential risk. However, if earthquakes, even of modest magnitude, can damage the caprock and jeopardize its sealing capacity, CCS may result in an unsuccessful strategy for significantly reducing greenhouse gas emissions (Zoback and Gorelick, 2012).
Fault internal structure, mechanical properties and fluid flow are inextricably coupled (Caine et al., 1996; Faulkner et al., 2010). Despite fault zone complexity, a close coupling exists between thermal, hydraulic, mechanical, and chemical processes in fractures and faults. Several studies have focused on understanding the geomechanical processes related to $CO_2$ injection, at both the lab and field scale (Vilarrasa et al., 2019 and references herein). Often caprock failure has been linked to changing thermo-hydro-mechanical (THM) processes. Numerical simulations have highlighted how fault/fracture reactivation in
caprocks are affected by two-phase fluid flow (Jha and Juanes, 2014), the presence of heterogeneities (Rinaldi et al., 2014), or temperature changes (Vilarrasa et al., 2017). Field studies on deformation (Vasco et al., 2010; Vasco et al., 2018) corroborated by numerical modelling (Gemmer et al., 2012; Shi et al., 2013; Rinaldi & Rutqvist, 2013; Rinaldi et al., 2017), have highlighted the importance of potential caprock failure on successful $CO_2$ storage. The role of chemical-mechanical processes on fault reactivation have been researched in the laboratory (Le Guen et al., 2007; Hangx et al., 2013; Amann et al, 2017; Vialle &
Vanorio, 2011; Mikhaltsevitch et al., 2014) and with field evidence (Rinehart et al., 2016; Hovorka et al, 2013; Al Hosni et al., 2016), mostly for storage reservoir rocks. These geochemical reactions could influence other characteristics of the rocks (e.g., mechanical parameters), which can in turn be linked to fault reactivation and induced seismicity (Vilarrasa & Makhnenko, 2017). A limited set of experimental studies have reported on the chemical processes that occur in caprocks, such as clays, shales and carbonate rich shales (Kaszuba et al., 2005; Credoz et al., 2009; Alemu et al., 2011; Kapman et al., 2016),
presenting a relevant knowledge gap when it comes to successful $CO_2$ storage.

A large amount of empirical observations is provided by $CO_2$ injection operations conducted by oil and gas industry (e.g. Jia et al., 2019, Michael et al., 2010 and references herein), but experiments targeting faults, especially on geomechanical and





geochemical coupling are still quite rare. Over the past decade a few in situ-scale experiments have been conducted on controlled release of $CO_2$ (free phase or dissolved) to better understand environmental impacts and test monitoring techniques

(Roberts & Stalker, 2017, and references therein). The experiments differ in many aspects, such as geological environments, injection depth and injection strategy. Most experiments released $CO_2$ into unconsolidated formations such as sand or gravel with only one exception to date, where gaseous $CO_2$ was injected into a lithified carbonate formation at only 3 m depth (Rillard et al., 2015). These experiments usually mimic the effects of leakage from wells by injecting into a vertical structure from a point source. The leakage through a fault is simulated by injecting from a linear feature. Besides the experiment described in

this contribution, to our knowledge at date, only one other test-site, the CISIRO In-Situ Laboratory Project in Western Australia, aims at targeting a fault in a reservoir formation with the injection of a small volume of $CO_2$, with the purpose to evaluate the ability to monitor and detect unwanted leakage of carbon dioxide from a storage complex (Michael et al., 2019). The Australian experiment is ongoing as injection started in early 2019 (K. Michael, pers. comm.).

Recently the Mont Terri rock laboratory (MTRL) hosted a decameter scale experiment (FS experiment), aiming at observing

the rupture and sliding mechanisms on a fault subjected to injections of large amounts of fluid. The FS experiment aims at understanding the conditions for slip activation and stability of clay faults, and the evolution of the coupling between fault slip, pore pressure and fluids migration. The experiment revealed complex rupture mechanism associated to micro seismicity (e.g. Guglielmi et al. 2020a, b). Results obtained by the experiment are crucial in defining mechanisms of natural and induced earthquakes, their precursors and risk assessment. The FS test and its findings constitutes a valid basis to develop the

experiment we describe in this work.

However, experiments that aim at investigating the transport/migration of $CO_2$ in caprock formations at decameter scale under controlled conditions (i.e., confining pressure, pore pressure, temperature, saturation degree, etc.) are still rare. To our knowledge, the only experiment to date dealing with an injection prolonged for many months was conducted at Daniel Electric Generating Plant,, Mississippi, and specifically targeted the effects of dissolved $CO_2$ on shallow groundwater reservoir, and

not on the transport/migration of $CO_2$. Hence, the testing of faulted caprocks subjected to $CO_2$ injection and the monitoring of coupled geomechanical and geochemical effects is of particular interest, because it offers direct observations that will help to gain insight into coupled THMC processes in faulted caprock and ultimately to improve site assessments of storage sites.

We are currently running an experiment that aims at covering this knowledge gap by providing observations on $CO_2$ migration in a fault system, at a decameter scale, therefore under well-controlled conditions, but targeting a rock volume that can capture

heterogeneities representative of a large scale in-situ injection. We want to simulate a situation in which the $CO_2$ contained in a storage site reaches a caprock that is crosscut by faults. We want to target one of the most critical conditions, when $CO_2$ could escape the reservoir through the fault, and possibly contaminate fresh water aquifers in the overburden or/and reach the surface and release to the atmosphere (Fig. 1). The MTRL located in northwester Switzerland (Fig. 2), allows in-situ access to a fault ( Mont Terri Main Fault) hosted in a clay formation, the Opalinus Clay, and offers a unique opportunity for a prolonged





(multiple months) decameter-scale $CO_2$ injection experiment (CS-D: Carbon Sequestration - Series D) into a fault to study relevant geomechanical and geochemical processes on leakage and fault properties.

With the CS-D experiment, we want to understand how exposure to $CO_2$ affects the sealing integrity of caprock hosting a fault, through observation on permeability changes, on fluid migration along the fault, and on interaction with the surrounding environment. With this we want to test the hypothesis that the retention capacity of the rock, even if faulted, are not affected

by exposure to $CO_2$ if the pressure conditions are not exceeding the rupture of the fault. We also want to test the hypothesis of self-sealing of the fault after eventual rupture and slip events. We want to investigate to which extend it can be proved generalized the concerns expressed by Zoback and Gorelick (2012) that the sealing integrity of $CO_2$ repositories can be threatened by earthquakes even of small size. Indeed, not only the rock volume targeted by our experiment but also the duration of $CO_2$ injection is an element making our experiment unique.

Finally, we want to validate instrumentation and methods for monitoring and imaging fluid transport, we want to generate well-constrained parameters as inputs for Hydro-Mechanical-Chemical (HMC) simulations and validate the observation on the mobility of $CO_2$ in the caprock and the possibility of leakage in the overburden.

The general concept of the CS-D experiment is to introduce $CO_2$-saturated water and tracers in the fault at MTRL, in a long-term (12 months) steady state continuous injection, forerun and intercalated by short, pulse-wise, pressure increase steps that

would be repeated at regular intervals during the long-term injection. The effects of the injection are monitored by synchronized monitoring system with the aim to observe changes in pressure, strain, and seismic velocity in the rock volume surrounding the injection site. Moreover, observations on fluids at a site nearby the injection would allow for detection of tracers. Microseismicity is also monitored during the pulse test operations.

Lab scale mechanical characterization on rock samples, geochemical fluid sampling, collected at the injection site, and

numerical simulations on the hydraulic response of the fault (pre-, during-, and post-injection) support the in-situ observation and offer key parameters for their interpretation.

This paper presents the concept of the CS-D, a general overview of the CS-D test-site, the conceptual design and the details of the experimental instrumentation. We also present results from the characterization of the rock volume prior to $CO_2$ injection, which provided the parameters adopted for the continuous long-term $CO_2$ injection that is currently ongoing. Finally, we

discuss the implication of the current observations, and speculate on the potential impact of the long-term experiment.

## 2 Overview of the Experiment

### 2.1 Aims

With the CS-D experiment, we aim at better understanding the processes governing:

1. the mobility of $CO_2$-rich water through the damaged zones of a fault;





2.   the impact of long-term (~ca. twelve months) exposure to $CO_2$-rich water on the permeability/porosity in the damaged zone and in the intact rock;

3.   the coupled geochemical and geomechanical variation due to rock-water interaction;

4.   the propagation of the transient field pressure in the fault and in the host-rock;

5.   the deformation of the rock mass as response to pressurization and slip, if any;

6.   the occurrence of induced micro-seismicity in clay.

With its dense network of monitoring systems, the experiment aims at collecting multiparameter observations and data from independent but strongly integrated monitoring techniques. Over the course of the experiment, we aim to establish a dataset at high spatial resolution that will yield insight into the interrelationship of hydraulic, geomechanical, and geochemical processes within the fault. Therefore, a dense network of multi-parameter sensors is installed around the injection site, to monitor the rock volume hosting the fault zone. The installation of the CS-D experiment in MTRL includes micro-seismicity and active seismic monitoring, cross-hole electrical resistivity monitoring, axial deformation, coupled with a with a three-dimensional displacement probe (SIMFIP, Guglielmi et al., 2013), and in-situ dissolved gases monitoring. The installation of a permanent infrastructure allows for long-term experiments.

Direct laboratory measurements on rock and fluid samples at centimeter scale are also an essential part of the experiment. The aim is to integrate the geophysical observation with analytical tests both on fluid samples collected within the fault and on rock samples either from the fault or from the intact rock. Samples from intact rock and fault zone collected before and after the injections will allow characterization of mineralogical and chemical changes due to the rock-fluid interaction; geomechanical tests will reveal eventual changes due to exposure to $CO_2$-rich water. This part is not discussed in the present paper and a detailed description of the methodology adopted for lab studies is in Wenning et al., 2019.

Numerical modelling assists the design and analysis of different phases of the experiment and a primary aim of the experiment is also to provide parameters to validate and calibrate numerical models that can be used for enhancing process understanding, sensitivity studies and upscaling.

The CS-D experiment takes advantage from close cooperation with a partner experiment, FS-B (Guglielmi et al., 2019), at a nearby location, and with complementary objectives. Both experiments deal with fluid injections into faults, but the main aims of the experiments are different:

-   FS-B's main aim is to imaging fluid flow, permeability and stress variations during rupture along the Main Fault zone, to better understand the role of fluids in earthquake rupture and fault reactivation. The experiment will feature large volume injection, and long-term monitoring of the recovery phase.





- CS-D experiment focus on the role of $CO_2$ on hydro-mechanical-chemical characteristics properties and addressing
the long-term behaviour of clay to several months of low flow rate injection.

The time schedule of the two projects is closely coordinated. Being the FS-B experiment a stimulation of the same fault targeted by the CS-D injection, it simulate the case of a seismic event causing a rupture and slip, and giving the opportunity to the CS-D motoring equipment to record the event and to detect possible leakages due to the slip. The FS-B stimulation is planned at a late stage of the CS-D experiment, after a long-term injection phase in CS-D, when observation on the evolution of the $CO_2$
in the fault indicate a steady-state flow path.

## 2.2 Location

The MTRL is an underground facility located in north-western Switzerland (Fig. 2a) in the Jura Mountains, a small fold-and-thrust belt (Fig. 2c) that represents the youngest and most external deformation zone of the Alps (Pfiffner, 2014). The laboratory is located 280 m below the surface and comprises c.a. 700m of galleries and niches.. The MTRL, under the
responsibility of the Swiss Geological Survey, swisstopo, hosts experiments of twenty-two organizations from various countries worldwide, and offers a technical and scientific platform facilitating the realization of scientific projects in the field of deep geological disposal.

Experiments in the MTRL investigate the properties a pristine claystone, the Opalinus Clay, that has been indicated as a possible host rock for radioactive waste in Switzerland (Bossart et al., 2017 and references therein). Because of its very low
hydraulic permeability and its regional extension over the Swiss Molasse basin, the formation is also considered a good seal for underground reservoirs (i.e., $CO_2$ storage) at the regional scale.

The Opalinus Clay, deposited around 174 Ma ago (Hostettler et al. 2017), is sequence of shales, that Based on the content of clayey minerals, quartz and carbonate, three main facies have been defined: clayey, sandy and carbonate-rich facies (Blaesi et al. 1996; Hostettler et al. 2017) . It is an overconsolidated clay, which has reached an estimated maximum burial depth of
around 1350 m at the current laboratory area, with an overconsolidation ratio of almost 5, assuming a current average overload of 280 m (Hostettler et al. 2017). The formation is 131 m thick (Bossart et al., 2017) at the MTRL. The Opalinus Clay and the adjacent formations form an anticline, crosscut by the tunnels of the lab. This structure is interpreted as a fault-bend fold, where a series of  thrust faults, extensional faults, and strike-slip faults intersect in a complex pattern (Nussbaum et al. 2011). Among these tectonic features, the most evident is the so called Main Fault, a thrust fault, located in the shaly-facies, with
shear movement towards NNW (Nussbaum et al., 2011). The Fault zone consists of several architectonic elements: fault gouge, S–C bands, meso-and micro-scale folds, numerous intersecting fault planes, and apparently undeformed volumes. The thickness of the fault zone varies between 1 and 4.5 m (Nussbaum et al., 2011).



The CS-D experiment takes place across the Main Fault, accessed from the newly excavated niche 8 (Fig. 2 b). The niche is located entirely in the shaly-facies of the Opalinus Clay. A set of vertical and inclined boreholes have been drilled and

instrumented from niche 8 going through the fault zone.

**2.3 Timeline**

The CS-D experiment comprises three main phases (Fig. 3).

- **Phase 1**: preparation of the long-term injection phase. It included the planning, and realization of the experimental infrastructure: drilling of boreholes, logging, collection and storage of samples, boreholes completion and

instrumentation, installation of surface geophysical monitoring equipment and all ancillary operations aimed at completion of the experimental infrastructure. Moreover, in this phase we characterized the rock volume in terms of geological and structural features, hydraulic properties, and seismic velocity distribution, all of which are essential for the design of the long-term fluid injection and the interpretation of experimental results. During the first phase, one main goal was to define the long-term injection pressure. To this end we performed repeated hydraulic tests in

order to define the pressure at which the fault suddenly show a large increase of flow rate for small pressure increments, other-wise known as the Fault Opening Pressure (FOP, Guglielmi et al.2016). The concept of FOP, defined as the pressure threshold at which a large increase in flow rate is registered at the injection point, and considered the start of activation of the fault (Guglielmi et al., 2016) was particularly helpful in designing the CS-D injection protocol.  We performed these tests in different injection intervals within the fault in order to determine

what, if any, zones are more reactive to pressure changes, and plan the installation of fluid sampling and monitoring equipment. The experimental installation was performed from August to December 2018. Baseline acquisition started immediately after the installation completion and lasted until May 2019. The results of these baseline measurements are presented in this paper.

- **Phase 2**: long steady state injection, below the FOP. We inject $CO_2$-saturated water and tracers while we keep

monitoring for pressure, electrical resistivity and pH, seismic velocity changes, and gas content. Water is sampled from the fault at regular time intervals and analysed in laboratories to monitor for chemical variations. In this phase, we will repeat regularly hydraulic tests and compare the results to detect possible variation in the fluid mobility. The long- term constant pressure injection with $CO_2$-saturated water started in June 2019 and is planned to last c.a. twelve months. This phase will be terminated when the neighbour experiment FS-B will take place. The FS-B will inject

large quantities of water at high pressure at a distance of c.a. 20 m from CS-D injection point, aiming at substantially increasing the permeability of the fault zone.





- **Phase 3**: post long-term injection operations. We will repeat characterization tests and prolonged injection test to identify possible effects of the FS-B experiment on the CS-D injection site. This phase also may include the collection of rocks cores from sampling drills that will reach the volume exposed to $CO_2$-saturated water, for further detailed

petrophysical and geo-mechanical, and geochemical characterization at the lab scale. A new long-term injection phase is foreseen to highlight possible changes in leakage rate after a major fault zone stimulation.

All the phases are integrated by continuous monitoring for pressure, deformation, pH and water electrical conductivity, as described in detail in the following chapters.

## 3 Layout of the installation

The layout of the boreholes used in the CS-D experiment was determined from constraints defined by the geometry of the fault, by bedding of the clay and by preliminary numerical modelling of fluid flow, to define the position of the boreholes and ensure accuracy of the monitoring instrumentation. To avoid interference from the excavation damage zone (EDZ) around the tunnel, we planned the injection boreholes and the fluid monitoring borehole to intersect the fault at a depth greater than 10 m, given the tunnel radius of 5 m (Bossart et al., 2002).

The position and geometry of the Main Fault was the basis to start the planning of the experimental layout. The depth of the fault was preliminarily calculated through the 3D geological model, provided by swisstopo. The Main Fault is primarily strike oriented N80° and dipping ~50-65 °SE. From the fault geometry model, we expected the fault to be approximately 10 to 25 meters below niche 8, increasing in depth towards the southern end of the niche, well below the EDZ (Fig. 5a). During the drilling phase, core mapping and borehole image logs provided more precise understanding of the Main Fault geometry and

allowed for adaptation to the borehole positioning.

### 3.1 Numerical scoping calculations

Preliminary numerical modelling results with the coupled numerical simulator TOUGH-FLAC (Rutqvist, 2011; Rinaldi & Rutqvist, 2019) helped constrain the distance between injection and monitoring boreholes. Such simulator allows solving for coupled fluid flow and geomechanics. For planning, we tested scenarios with both constant and stress-dependent permeability.

Similar to previous numerical modelling of the FS experiment (Guglielmi et al., 2020a,b), the injection occurs with constant head at the centre of a 20×20×20 m domain and with a planar feature representing the fault zone (Fig. 4a). The fault plane, embedded in a 3D model, is simulated with a finite width (1 m). The preliminary model assumed an injection strategy similar to the previous FS-experiment (Guglielmi et al., 2020), where the injection time was extended to account for a long-term injection (twelve months, given some preliminary planning of the experiment – Fig. 4b). The main goal of this preliminary

modelling was to assess the maximum reach of the pressure front and injected water.





In a first set of simulations, we assumed that the permeability remains constant during injection. We considered permeability values ranging from $10^{-21}$ to $10^{-19}$ m²; therefore, we simulated cases slightly more and slightly less permeable than as indicated in literature (Marschall et al., 2003-2005). Results show that the pressure perturbation has a long reach, while the injected water is confined in a few meters around the injection well. Figures 4c-d show the results for the case of permeability $5 \cdot 10^{-20}$

m²: while the pressure front can reach up to 8 m distance, the injected fluid is confined around the injection point. A larger/smaller permeability results in larger/smaller reach of the injected fluid. Simulating permeability varying as a function of elastic or tensile opening of fracture results in a much larger reach as soon as the pressure is increased above the leakage threshold (Zappone et al., 2018). However, this condition was considered a worst-case scenario, as the CS-D experiments aims at injecting at a pressure below the FOP.

Results of the preliminary modelling suggested that a fluid monitoring borehole needed to be placed at the minimum distance given the setup (i.e. 2 m inter-distance at gallery floor). All the different scenario results with fixed permeability and stress-dependent permeability are provided in the Supplementary material ( S1 and S2, respectively).

### 3.2 Boreholes geometry

Due to the considerable anisotropy of seismic (e.g. Nicollin et al. 2008; anisotropy=28%) and electrical (e.g. Nicollin et al.

2010; anisotropy=85%) properties of Opalinus clay, the geophysical monitoring boreholes were oriented in a way which facilitates the data processing. For this reason, they are oriented such that the 2-D tomographic planes in between these boreholes are oriented normal to bedding planes (anisotropy symmetry axis within tomography planes). Three geophysical boreholes were drilled with an inclination from vertical of 48-51° to have a perpendicular intersection with the fault and bedding. In addition, the intervals in the injection borehole needed to be in the vicinity of the field of view of the geophysical

arrays.

A total of seven boreholes, four vertical (BCS-D1, -D2, -D6, and -D7) and three inclined (BCS-D3, -D4, and -D5), were drilled and equipped in order to perform and monitor a long-term (twelve months) injection of $CO_2$-saturated water in the fault, to monitor the movement of the water in the fault by geophysical methods and by sampling of fluids. The geometrical layout of the boreholes is illustrated in Fig. 5. Table 1 gives specifications on borehole purpose, diameter, length and orientation.

The boreholes BCS-D3 to -D6 were drilled for specific geophysical investigations. BCS-D3 and -D4 were drilled coplanar to each other, such that they create a plane that intersects the fault zone at a distance of 2 m from the injection intervals. BCS-D5 was drilled on the opposite side of the injection borehole and is also coplanar to BCS-D3. This allows for an additional tomography plane across the injection zone. BCS-D6 is drilled vertical at a distance of 3 meters form BCS-D7 with the purpose of placing sensors for continuous seismic monitoring for better locating seismic events in 3-D.





### 3.3 Instrumentation

#### 3.3.1 Injection system

The injection system is designed to perform injection separately at four depth intervals in the fault zone. An injection module (Fig. 6) designed for injection over long periods was connected to the borehole BCS-D1. The injection is done via a syringe pump (Teledyne ISCO 500D), that is remotely controlled through a dedicated software (DCAM) developed by Solexperts AG. The pump allows the injection of moderate volumes of fluids, with an injection chamber volume of 0.5 l, refill breaks of 2.5 min, and has an accurate control of injection rate from 0.001 to 200 ml/min. The pump is connected to a 10 l tank, where the injection water is pressurized at 2 MPa and is mixed with $CO_2$ and Kr by bubbling. A circulation pump and a flow meter enable controlled mixing. The injection module is designed for permanent and remote-controlled injections.

#### 3.3.2 Boreholes instrumentation

Borehole BCS-D1, the injection borehole, and BCS-D2, the fluid-monitoring borehole were equipped with a 4-fold and 6-fold packer system, respectively. Figure 9a displays a schematic layout of the packer system showing packer and interval depths and the intersection with the fault, as from borehole logging.

Pressure at each injection and monitoring interval is monitored with sensors connected at the surface; flow lines and packer lines are realized in stainless steel to avoid corrosion from $CO_2$ exposure.

Deformation is monitored through Distributed Strain Sensing (DSS) fiber optic (FO) cables, integrated in the packer system, and anchored at each interval to avoid effect of packer inflation on strain measurement. The multiple packers system allow for multiple injection/monitoring intervals, both in the fault zone and in the host rock. To resemble the natural composition of formation water, all the intervals were saturated with Pearson water A1 type (Marceau et al., 2016), depleted of Mg and Ca, to avoid mineral precipitation due to $CO_2$ and consequent clogging of the lines.

Boreholes BCS-D3 to 6 were all cased with PVC tubes to assure impermeability of the inner chamber and instrumented with DSS FO cables fixed to the casing to allow distributed deformation measurements. After installation, the annulus between the PVC casing and the borehole wall was grouted with a mixture of bentonite and cement. In order to prevent any possible leakage of $CO_2$ at the surface of the niche through the monitoring wells, epoxy resin was injected between PVC and borehole wall at depth below the EDZ and above the fault.

The fiber optic strain sensing cable, 3.2 mm in diameter, are flexible cables, armoured with a central metal tube surrounded by a structured PA outer sheath, containing one single optical fiber (BRUsens strain sensing cables). They are designed to measure strain range up to 1% (10000 μstrain). Axial deformation is also measured with a chain potentiometer grouted outside the PVC casing in borehole BCS-D5, including 12 measuring sections, ten of which are crossing the fault with 0.5 m inter-distance between each element. The chain potentiometer consists of anchor-elements connected to each other by PVC tubes. The



anchors measure unidirectional displacements relative to each other. Standard potentiometric displacement sensors with a
      measuring range of 100 mm are used for the measurements in the chain.

      Fifty ring-shaped stainless-steel electrodes, with an interspacing of 0.5 m, were clamped to the casing in BCS-D3 and 4, to
      allow electrical resistivity tomography and time-lapse observations on a 2-D plane between the two boreholes. Inside the
      casing of BCS-D3 a 3-component geophone array of twenty-four geophones (100 Hz) with interspacing of 0.5 m was installed,
while identical single 3-component, geophones were installed at the bottom of boreholes BCS-D4 to 6. The 3-component
      geophones were custom-designed by Omniquest Int. and are identical to the ones installed in a previous MTRL experiment
      (Manukyan and Maurer, 2018). Eight piezo-sensor elements for high frequency seismic detections were installed in boreholes
      BCS-D5 and 6 (four each).  We used piezoelectric sensors (type GMuG MA-Bls-7-70) designed by the Gesellschaft für
      Materialprüfung und Geophysik (GMuG). These sensors are similar to those commonly used in laboratory acoustic emission
experiments (e.g., Ishida, 2001) and are highly sensitive in the frequency range of 1–100 kHz, with the highest sensitivity at
      70 kHz.

      As seismic sources, a P- and S-wave sparker is employed in the water-filled PVC-casings of borehole BCS-D4 and BCS-D5
      (in D5 for baseline and optionally after the injection experiment). Both these boreholes are in-plane with borehole BCS-D3,
      enabling 2D tomography within the two corresponding planes in relatively high resolution.

### 3.3.3 Seismic instrumentation in the niche 8

      To complete the seismic monitoring network (active and passive), thirty-three one-component geophones (Geospace Corp
      Corp. Texas) were coupled to the rock behind the shotcrete in the niche. An additional network of 18 piezo-sensors (Type
      GMuG Ma-BSL-7-70 sensitivity between 1 and 100 kHz) were set up at the surface (clamped to the niche floor), in two parallel
      lines along the tunnel walls. The sensor spacing is about 2 m. As the piezo-sensors do not have a well-defined instrument
response due to resonance peaks that depend upon sensor design and local installation to the rock (Kwiatek et al., 2011), we
      combined one piezo-sensor with a calibrated one-component accelerometer (type Wilcoxon 736T) that has a flat instrument
      response in the range 2–17 kHz. The surface instrumentation also comprises six hammer sources installed in a parallel line
      along the tunnel wall and are within the plane between BCS-D3 and D4.

### 3.3.4 Fluid sampling and dissolved gas sampling

Two circulation modules were installed for fluid sampling in the intervals of the 6-fold packer system in borehole BCS-D2.
      The first circulation loop is connected to a gear pump with a flow meter that allows fluid circulation in one selected interval
      from which fluid samples can be extracted and collected in stainless steel vials maintaining in situ pressure conditions, and
      therefore avoiding degassing. The circulation prevents chemical precipitation in the interval. An EC probe (Hamilton
      Conducell, 1 – 300 mS/cm) and a pH probe (Hamilton Polilyte Plus, pH range 0-14) are also connected in flow-through





cell. Fluid major element compositions are determined using ion chromatography. Carbon isotopes in dissolved inorganic carbon are determined by isotope ratio mass spectrometry after acidification of the sample to pH < 2 to quantitatively extract the inorganic carbon as $CO_2$ with an analytical reproducibility better than 0.2 ‰.

A second circulation loop (Fig. 7) allows for *in-situ* analysis of dissolved gas in the fluids in a selected interval. The measurements are performed with a portable mass spectrometer called "miniRuedi" (Brennwald et al., 2016) that allows

quantification of He, Ar, Kr, $N_2$, $O_2$, and $CO_2$ partial pressures with an analytical uncertainty of about 3%. In addition to the monitoring $CO_2$, we seek at tracking the restitution of Kr that is used as artificial conservative tracer. In this perspective, both conservative and reactive transport processes will be quantified and discussed. We also focus our attention on the evolution of natural dissolved noble gases such as dissolved He and Ar that might reveal mixing with in-situ fluid remobilization (Roques et al., 2020).

### 3.3.5 Pressure/Displacements/Water Resistivity monitoring in boreholes CS-D7

The CS-D experiment uses a new prototype of the SIMFIP hydromechanical borehole probe (Guglielmi et al., 2013). This new SIMFIP prototype, developed at the Lawrence Berkeley National Laboratory, has been designed for long-term borehole monitoring of micrometre fault zones displacements and eventual associated $CO_2$ leakage. Main prototype development concerned the possibility for a SIMFIP probe to isolate a long borehole interval including an entire fault zone thickness that

would have been identified from an initial borehole logging program. The challenge was to harden the micrometre resolution of the displacement measurements over a decametre long device and over a several months-to-years monitoring period. The MTRL, its Main Fault and the CSD project are the perfect site to test this new prototype at the relevant field scale.

In order to monitor the hydromechanical behaviour of the entire fault zone, a 6.3m long SIMFIP interval has been designed, sealed by two 0.9 m long inflatable packers. In this configuration, the SIMFIP sensor is measuring the relative displacement

of the upper packer, the lower packer considered fixed. The packers play two roles, sealing the interval to isolate fault zone pore pressure variations and anchoring the SIMFIP sensor to measure the displacement of the fault hanging-wall relative to the foot-wall. A compass set above the upper packer allows orienting the displacement measurements. The SIMFIP sensor is a 0.49m long and 0.1m diameter pre-calibrated aluminium cage set on the tube connecting the two packers. When the fault straddled by the packers' interval is deforming, the cage allows obtaining angle dependent strain measurements which can be

used to constrain the full three-dimensional displacement tensor and the three rotations of the upper packer relative to the lower packer.

Borehole pressures are monitored below the lower packer, between the packers and above the upper packer. Water resistivity electrodes have been distributed every 5.54 cm along the entire length of the SIMFIP chamber in order to localize where eventual leaks could occur from the fault zone into the borehole. It was assumed that for example dissolved $CO_2$ leak would

slightly change the formation water resistivity, enough to be detected by the resistivity probe. After several months of tests



and monitoring, measurement sensitivities are of $10^{-6}$ m for displacements and $10^{-3}$ Pa for pore pressure (Paragraph 4.4). A remote control allows programming the SIMFIP, and for example vary the sampling rate from 1 Hz to 1 kHz depending on the testing protocol.

## 4. Results from Phase 1

### 380  4. 1 Main Fault geometry from cores and logs

All boreholes were cored with exception of BCS-D4. Cores were recovered with a double barrel technique, with the exception of D1 were the Main fault section has been cored with a triple barrel technique. Core recovery was higher than 90%. The boreholes were logged with an oriented optical borehole televiewer, total count natural gamma ray, dual induction, and calliper (four arms in vertical single, arm in inclined boreholes). A cross-analysis of the cores and of the logs allowed an accurate

reconstruction of the oriented geological log for each borehole. The position of the instrumentation in injection and fluid monitoring boreholes was then decided, based on the fault depth.

The bedding is uniform on both sides of the fault with a mean orientation striking N053 ° and a dip of 46° SE. A top and a base plane, both clearly identified on cores and image logs (Fig. 8a, b), spaced c.a. 1.5 to 3 m, define the Main Fault. In the core there is a sharp discontinuity between undeformed bedding and highly deformed scaly clay material, that consists of a

tangled web of slickensides, bounding largely unaltered microlithons (Jaeggi et al., 2017), and is also visible in the image logs. Table 2 displays the depth and orientation of the top and bottom of the Main Fault in all boreholes from image logs analysis. The top and bottom of the Main Fault plane within our study area are variable, with typical strike oriented N031-068° and dipping 56-65°SE. In borehole BCS-D5, the top of the fault has a very steep dip, which is near vertical or perhaps overturned. The fault material between these two planes is heterogeneous, including zones with fault gouge, C'-type shear

bands, meso-scale folds, microfolds, numerous fault planes, apparently undisturbed parts and a 'scaly' fabric where the rock splits progressively into smaller fish-like flakes. Moreover, sparse and discontinuous secondary fault planes have been observed above and below the Main Fault.

There are numerous fractures and faults within the Main Fault. The location of the injection (BCS-D1) and monitoring (BCS-D2) intervals within the Main Fault are depicted in Fig. 8c. Within the injection borehole there are fractures with an orientation

similar to the Main Fault, as well as a set with a conjugate orientation (Fig. 8b, d). Borehole injection tests revealed a pressure response in monitoring interval M1 and M2 (see nomenclature of intervals in Fig. 8c) when injecting from injection interval Q4. As such, the structures within these intervals are highlighted in Fig. 8d. While we do observe northward dipping fractures that might connected the Q4 and M1/M2 intervals in other parts of the borehole, these fractures are not directly observed in these intervals.





Core pieces longer than 10 cm were sampled and sealed after undergoing an on-site quality control (i.e., no open fractures, drilling induced features, etc.). Samples were wrapped in barrier foil aluminium laminate and vacuum-sealed to limit contact with air. Samples were labelled with respect to the borehole name and order number. For samples that contain fault zone or carbonate lenses, this property is also written beside the sample name on the label. After sealing and labelling samples were stored in a wooden box and shipped to ETH Zurich. Studies on multi-flow transport in fractured rock and geomechanical characterization of the fault and host rock have been developed in laboratories respectively at Imperial College London and ETH, and at EPFL.

## 4.2 Hydraulic characterization

We performed several injection tests to estimate hydraulic properties and opening pressure (FOP) of the fault zone. These tests were essential to identify which intervals would be best candidates for injection and monitoring. Injection always occurred in intervals of the BCS-D1 borehole. The different tests are summarized in Table 3 and include:

- Long step test at constant head with 28-30 hours inter-step time (LST).
- Short step test at constant head with 5-10 minutes inter-step time (SST).
- High pressure short step up test at constant head with 5-10 minutes inter-step time (HP-SST) these tests were usually performed after a LST.
- Pulse step test with pump stopped after reaching desired pressure then 10 minutes decay before new step (PST).

While we performed the tests in several intervals (Q1, Q2, Q4), we dedicate our analysis only to interval Q4, which is the shallowest interval in the injection borehole (BCS-D1), which is the one finally chosen for the $CO_2$-saturated water injection in Phase 2. Our tests gave first order estimates of the initial transmissivity of the fault zone from this interval. Figure 9 shows the results of a LST in interval Q4: only when injecting in this interval we observed a hydraulic response in the fluid monitoring borehole (BCS-D2). This was a critical information to decide the injection interval for the Phase 2 injection. Figure 5a (insert) and 9a shows a schematic (plane view) of the distance between the interval Q4 and two monitoring intervals (M1 and M5). Quite interestingly, when injecting from Q4, only the bottom intervals of the fluid monitoring boreholes showed pressure variation (e.g., M1 less than 0.1 MPa, Fig. 9b, red curve), while all the others show no variation (e.g., M4, Fig. 9b, green curve). We note that the flow rate never reaches steady conditions for all the steps performed in the analysed test (Fig. 9c).

The transmissivity of the fault zone was estimated by analysing a pulse test (still performed in interval Q4) with a Cooper-Bredehoeft-Papadopulos-Neuzil model (Cooper at al., 1967; Neuzil, 1982; Renard, 2017). Figure 10a-b showed the pressure variation and the fit with the model that results in a transmissivity of $1.8\times10^{-13}$ m²/s. In the model, the pressure decay is normalized to the peak, and both pressure variation and its derivative are used to estimate transmissivity. We discarded the first 30 seconds of data as these could be influenced by interval storage. For this model, the compressibility of the system was estimated given the injected volume to reach the peak pressure (1.7 MPa). A similar value of transmissivity ($2.8\times10^{-13}$ m²/s)





is obtained by analysing the first step of the LST (Fig. 10c-d) with a Jacob and Lohman model (Renard, 2017). For this model, only the flow rate is used to estimate the transmissivity. As above, we discarded the first seconds of data as these are strongly affect by the interval volume and near-borehole skin effects.

### 4.3 Fault opening pressure

The fault opening pressure (FOP) in the interval Q4 was determined prior to starting the long-term injection with $CO_2$-saturated water in Phase 2. During the PST-test, the pressure was increased in steps of 0.3 MPa and shut-in after reaching the desired pressure. Figure 11a shows the pressure recorded at injection interval (BCS-D1-Q4). Figure 11b shows an enlargement of the recorded pressure when the pressure drop was more consistent. Figure 11c shows the pressure difference after 10 minutes waiting time compared to the injection pressure. The pressure response is non-linear when injection pressure was raised above

4.5 MPa. The large steps (0.3 MPa) employed do not allow for a precise measure of the FOP, but we can conclude than it is in the range 4.5-4.8 MPa. We performed this test only in the injection interval Q4 because it was the only interval showing pressure response in the monitoring borehole. After reaching this "reactivation", we started the long-term $CO_2$-saturated fluid injection below the FOP at a constant pressure of 4.5 MPa. An analysis of the decay curve from 4.8 MPa with a Cooper-Neuzil model (Renard, 2017) results in an estimated transmissivity of $9.2 \times 10\text{-}12$ m2/s, which is more than one order magnitude

increase compared to previous estimates (see above). Worth to note that for the estimate, here we considered only the last step and analysed the decay from considering the previous step pressure (i.e. from 4.8 to 4.5 MPa).

### 4.4 Borehole stability monitoring in BCS-D7

The SIMFIP probe was installed in the borehole immediately after drilling on October 2018. Figure 12a shows the location of the probe across the fault zone. The installation phase was followed by a period of tuning the packers' pressure until December

19th. Figure 12b-d shows the results of the first 5 months monitoring period, with the packer testing phase highlighted by the red shaded area. The main issue was to maintain constant the packer pressure. For example, during November 2018, packers show a slow deflation that requires several manual re-inflations. The problem was fixed in late December 2018 by installing an automatic control of the packer pressure. The four periods of packer pressure variations of January 10th, 16th, February 1st and March 27th correspond to complementary, manual adjustments of the packer system. The control of the packer response

is crucial because this probe is equipped with sliding-end packers in order to ensure an optimal sealing of the isolated interval that permanently matches with borehole dimension evolution related to interval pressure variation and to borehole clay walls deformations. Thus, because the packers slide while their pressure is varying, it affects the chamber pressure and the displacement measurements (since the SIMFIP is anchored with the packers – Fig. 11d). The packers pressure increase is inducing a chamber pressure decrease, a SIMFIP vertical extension (positive Dz variation) and an equal radial displacement





(EW = -NS, see for example February 2019 – Fig. 12d). This response matches with laboratory calibrations, and thus any deviation from it observed in the field might highlight a true hydromechanical evolution of the formation.

Interestingly, the chamber pressure increased of 0.2 MPa during the first month after installation. This period (red shaded area in Fig. 11b-d) could be interpreted as a stress relaxation after borehole drilling. Then, from early December to February, pressure decreased, indicating a potential coupled hydromechanical relaxation (orange shaded area in Fig. 11b-d). Finally, the

pressure in the SIMFIP 'stabilize' at about the initial pressurization (~0.3 MPa) in February 2019 (green shaded area in Fig. 11b-d). These long-term pressure variations are not clearly related to packers' effect (although the influence of the packers is observed over shorter periods). Both variations thus relate to the complex borehole pressure equilibration with formation pressure, which occurred in about 5 months. Displacement variations followed these long-term pressure variations. Displacement amplitudes are of 0.3 to 1 mm, the norm of displacement vector in March 2019 being estimated to 0.95 mm after

5 months of monitoring (more than 70% of the displacement occurred after about 1.5 month). These values are in reasonable accordance with strain relaxation effects associated to borehole or gallery excavation observed in other Mt-Terri experiments (Amann et al., 2017). The SIMFIP data show displacement variations in all the three EW, NS and Z directions, highlighting the three-dimensional characteristics of such relaxation effects.

The pressure in the interval, pressure in the packer, and displacement have been overall constant since then, with some

interruption due to on-site operation in April 2019 (grey shaded area in Fig. 11b-d). Starting from April 2019, the effect of a nearby tunnel excavation are visible on the probe, up to a maximum displacement of about 0.2 mm (Rinaldi et al., 2020).

A detailed view of the relative and detrended displacement for the same period as the LST test in Fig. 9 highlight the accuracy of the SIMFIP of about 1 μm (Fig. 12e), while the accuracy improved less than 0.5 μm after better calibration of the packer pressure (Fig. 12f).

**4.5 Seismic characterization**

Active seismic baseline measurements were conducted in January 2019, before the first injection test. They were repeated on June 11, 2019, before the start of the long-term injection. These baseline measurements were performed with hammer sources employed on the gallery floor and with a seismic sparker source employed in borehole BCS-D4 and -D5, while all geophones and piezo-electric transducers installed in boreholes and in the gallery were recording. Example data are shown in Fig. 13.

They have been recorded with a geophone cemented in borehole BCS-D3, while the sparker source was fired in 25 cm intervals in borehole BCS-D4. It shows a clear P-wave arrival (A), which is slightly delayed for sources fired within the Main Fault (B). Additionally, slow- and fast S-wave modes can be identified (C) and strong linear events, which are caused by tube waves propagating along the source borehole (D) and are reflected at the Main Fault (F).

The data shown in Fig. 13 has been processed using the following processing steps:

-     Median filter





- Bandpass filter
- Zero-time correction by cross-correlation of recorded trigger signals
- Trace stacking of 5-10 repeated shots

For characterizing the seismic P-wave velocity ($V_P$) structure in the region of the CS-D experiment, 2-dimensional P-wave
travel time tomography was carried out within planes between boreholes BCS-D3 and -D4 as well as between boreholes BCS-D3 and -D5. Here we show the tomogram measured with a P-wave sparker source from borehole BCS-D4 and with the cemented geophone array in borehole BCS-D3. The tomographic imaging involved the following procedure:

- Automatic picking of first arrival times
- Picking refinement with cross-correlation (e.g. Schopper et al., 2020)
- Anisotropy correction
- Iterative travel time inversion (after Lanz et al., 1998).

The clay bedding, along which the $V_p$ attain maximum values, is oriented normal to the tomographic planes. Therefore, it was assumed that no off-plane effects occur due to the $V_P$ anisotropy. Anisotropic effects within the tomographic plane were minimized by normalizing the travel times. This travel time normalization is displayed in Fig. 14, where prior to normalization (a) $V_P$ along the bedding plane reaches values of approximately 2870 m/s and extrapolating to angles normal to the bedding planes yields $V_P \approx 2280$ (Fig. 14a). This leads to an overall $V_P$-anisotropy coefficient A = (VP,max–VP,min)/VP,min = 0.26. After normalization of travel times, normalized P-wave velocities $V_{P,n}$ are around 2580 m/s in average (Fig. 14-b).

In Fig. 15, the resulting $V_{P,n}$ tomogram between borehole BCS-D3 and D4 is shown. It was computed with the baseline-data recorded on June 11, 2019. The inverted $V_P$-model explains the observed travel times with an average RMS error of 0.05 ms. The velocity values of the tomogram are displayed as normalized values according to Fig. 14, which means that they are average values of the in fact anisotropic velocities. The location of the Main Fault, estimated by interpolating observations from all CS-D boreholes (Fig. 4a), is displayed by the thin black lines. The Main Fault is causing a clear low-velocity zone in the $V_P$-tomogram, which can be clearly identified by cross-hole seismic. Furthermore, $V_P$ appears to be higher in the footwall than in the hanging wall. Heterogeneities within these two units were most probably due to our assumption that the anisotropy is homogeneous, whereas in fact the anisotropy is slightly higher in the footwall than in the hanging wall as shown in Fig. 14a.

## 5. Discussion

Although focusing on the characterization of the reservoir and the development of our experimental setup at the meter scale, the initial results presented in this contribution already provide important insights for our understanding of the processes involved in large scale $CO_2$ storage operations.



The structural mapping during drilling allowed to adjust the boreholes position and to install the packers in the injection,
       monitoring, and SIMFIP boreholes to span the entire thickness of the fault zone. Core mapping and borehole optical televiewer
       logs clearly identify the Main Fault in the host Opalinus Clay , marked by cm-thick dark fault gouge and scaly clay texture
       (Jaeggi et al., 2017) . The contact between fault and host rock is sharp (Fig. 8a). The Main Fault in the vicinity of the CS-D
       experiment is between 1 to 3 m thick with typical strike oriented N031-068° and dipping 56-65°SE. The upper contact, marked
by the ~ 1 cm thick gouge layer followed by a ~ 10 to 20 cm thick scaly clay, is similar to the upper fault contact in Gallery
       98  (Jaeggi et al., 2017) and the faults observed in the FS-B boreholes (Guglielmi et al., 2020a). A layer of scaly clay also
       marks the bottom of the main fault.  These trends show the strike parallel similarities that span across the > 50 m in the rock
       laboratory. While the tops and bottoms have similar orientations to the Main Fault in the gallery and other boreholes
       (Nussbaum et al., 2011, Jaeggi et al., 2017, Guglielmi et al., 2020), the internal structure is very heterogeneous. The majority
of fractures within the Main Fault have a similar trend to the boundaries (cf. Jaeggi et al., 2017). However, several conjugate
       structures and fractures do not fall within the Main Fault trend. These fractures correspond to S-C and Riedel R- and P-
       structures (Nussbaum et al., 2011). The heterogeneity of structural fabrics within the fault core that characterize the Main Fault
       is a common feature of faults in clay and makes it difficult to determine the hydraulic properties of such faults. Internal
       architecture if a fault in clay has been described for example at the Tournemire underground lab in France (Dick et al., 2016,
and references herein) where the shale formation is crosscut by a km-scale,  0.2-2m thick fault. In this case a fault core has
       been distinguished from a damaged zone; the core comprises gouge in thin dark cm-thick bands, cataclasites, rock portions
       with folded foliation planes, and lenses of less deformed rock, all elements similar to the Main Fault at MTRL. The damage
       zones is represented by a dense network of small faults, fractures, and calcite veins, that extends 2-3 m from the fault core.
       Pulse tests in Tournemire revealed hydraulic conductivity along the core-damage zone boundary one-two orders of magnitude
greater the than in the undisturbed rock (Dick et al., 2016). Marschall et al. (2005) investigated hydraulically sections of the
       fault zone at MTRL, and showed that the permeability of faulted Opalinus Clay and of undisturbed rock is not significantly
       different.

       During Phase I, several injection tests were performed to characterize the hydraulic response of the site and to determine if
       there was any hydraulic response between the injection and monitoring boreholes. An important first result of the CS-D
experiment is that the injected fluid is channelized along preferential pathways rather than along the fault plane, although the
       transmissivity of these pathways remains extremely low (order of $10^{-13}$ m$^2$/s). During the hydraulic tests, a clear pressure
       response was observed in the monitoring borehole (BCS-D2, intervals M1, M2) when injection occurred in the upper most
       interval of the injection borehole (BCS-D1, interval Q4). If this observation were linked to poroelastic effect, we would expect
       similar pressure increase in all interval at approximately the same distance, although the heterogeneity of the medium might
generate some differences. As the pressure variation in the monitor borehole is only observed in the bottom intervals, we argue
       that the poroelasticity is negligible, and that the flow follows complex pathways within the fault. A pronounced fracture



possibly connecting injection and monitoring boreholes in the direction Q4-M1 was not identified in these specific intervals, but conjugate structures (NW dipping fractures, Fig. 8b, d) could explain the fluid pressure connection of the Q4 injection interval to the M1 and M2 monitoring intervals.


Fault transmissivity models assume that the system is a homogeneous porous medium, with radial flow form the borehole. However, it should be noted that the Main Fault may behave rather as a fractured system given the interconnection of specific intervals rather than uniform/homogeneous response. In other words, our estimate relies on the assumption of a homogeneous representation of the rock with effective hydraulic properties, which might fail in capturing the real pressure distribution in

such fractured environment. Nevertheless, the transmissivity estimates agree with previous tests conducted in boreholes reaching the fault in shaly-facies at ca. 9 m depth from another niche (Marschall et al., 2005) and could indicate a permeability in the fault zone in the order of $10^{-20}$ m$^2$, when assuming a layer as thick as the injection interval (1.4 m). An analysis of all other tests and decay curve results in similar value for the transmissivity.

The estimated permeability is extremely low, and extrapolating this value to full-scale injection plants would not results in

major $CO_2$ leakage if constant through the operational phase. Indeed, such a low value makes the fault as impermeable as the caprock itself, but changes in permeability and/or porosity due to geochemical/geomechanical process could in the long term affect the sealing capacity, albeit in a more heterogeneous way compared to what expected during planning phase of the experiment.

An additional injection test was designed to determine the fault opening pressure (FOP), i.e. the pressure at which the fractures

are jacked opened and allow leakage. By performing a series of step tests, the FOP occurred in the range 4.5 to 4.8 MPa. This value is in agreement with previous studies at Mont Terri (e.g. 5.4 MPa – Guglielmi et al., 2020a). While the recorded signal at the injection point clearly indicated a fracture reactivation, and the estimated transmissivity increased by one order of magnitude, the pressure response was not clear at the monitoring point. It should be noted however that the FOP test (PST3 – Table 3) was performed right after an important excavation that occurred in a near tunnel (Rinaldi et al., 2020), which could

have affected the local state of stress, allowing for opening of fractures/cracks at lower pressure.

The reactivation of a fracture resulted in a stronger pressure decay compared to previous steps, although no obvious deformation was recorded at either the SIMFIP, potentiometer chain, or fiber optic. It is then difficult to estimate the orientation of the reactivated fracture(s) by looking at the injection data alone. Some minor deformations were observed during previous tests in the period March/April 2019 in the potentiometer chain (see supplementary figure S3), but it is hard to discriminate

the small effect of injection from other processes occurring at depth in the long term, such as borehole stabilization or stress relaxation after excavation. For example, the SIMFIP probe, placed at about 7 m from injection point and being able to capture submicron deformation, should be able to record some signals. However, data show complex three-dimensional strain relaxation effects that lasted about 1.5 months after drilling. These effects with deformation in the order of microns are affecting



the boreholes' interval fluid pressures in and outside the fault zone, and may mask any effect linked to the injection. Such a "long" relaxation period is consistent with observations made in other Mt Terri experiments dedicated to the long-term hydromechanical behaviour of the excavation damage zone around galleries or boreholes in low permeable Opalinus Clay (Bossard et al., 2017). How permanent and how amplified are these effects in the fault zone will inform on the evolution of the damage zone in the near field of boreholes drilled through faulted caprocks.

From the active seismic baseline data recorded during Phase I, we analysed the P-wave velocity ($V_P$) anisotropy. $V_P$-values averaged over the entire rock volume including the Main Fault were observed to be around 2870 m/s in the direction of the clay bedding in the host rock, and around 2280 m/s normal to the clay bedding. 2-dimensional tomographic imaging was carried out using anisotropy-normalized P-wave travel times. The resulting tomogram was capable of clearly revealing the location of the Main Fault in the form of a pronounced low-velocity zone correlating well with direct borehole observations. Values of $V_P$ we observed in situ are clearly smaller than what has been previously measured by other researchers on drill cores in the laboratory (e.g. Bossart et al. (2017), 2220-3020 m/s normal to bedding, 3170-3650 m/s parallel to bedding), but exhibits a similar degree of anisotropy, with A = 0.26, which is close to A = 0.3 estimated after Bossart et al. (2017). Lower values in absolute $V_P$ compared to ultrasonic measurements has been expected in accordance with the Kramers-Kronig dispersion relation (e.g. Mavko et al., 2009). In situ seismic crosshole measurements have previously been performed by Schuster et al. (2017). For apparent velocities outside the EDZ, they recorded values of around 3100 m/s for ray paths approximately parallel to bedding, and around 2600 m/s for rays normal to bedding. Fitting an ellipse through the apparent velocities (similar to Fig. 14) they estimated an anisotropy coefficient A = 0.20, which is close to our observation. The difference could be due to heterogeneities within the shaly-facies of the Opalinus Clay, but also because in our case ray coverage normal to the bedding was poor, whereas Schuster et al. (2017) did not cover ray paths exactly parallel to bedding. To our knowledge, seismic traveltime tomography across a larger fault in Opalinus Clay hasn't been performed before, but Jaeggi et al. (2017) measured ultrasonic interval velocities along a borehole which is crossing the Main Fault in Mont Terri and is oriented around 40° to the bedding planes. Within Main Fault-sections with scaly clay, they observed distinctly reduced $V_P$ values as low as 2000 m/s. This is even lower than what we observed in Fig. 15, and can be attributed to the higher resolution of the interval velocities which enables to resolve individual sections of scaly clay, whereas with our crosshole measurements we measured $V_P$ of the Main Fault consisting of a mixture of scaly clay and lenses of undisturbed Opalinus Clay.

All the injection tests during the Phase I were also monitored for induced acoustic emissions. Eight piezo sensors installed in boreholes at a distance varying from about 2.5 m to 10 m from injection point (Fig. 5 a,b) were recording at very high sample frequency (0.2 MHz) during the all injection tests. Despite reaching elevated pressure (e.g. 6 MPa in test Q4 - HPP-SST – Table 4), and while the pressure response was clear at a distance of about 2.5 m, no signal was recorded by the piezo-sensors.





This is not surprising, given the very low tendency of clay rock to generate seismic events (Orellana et al., 2018), but in similar conditions some seismicity was observed at a near experiment with resulting much enhanced flow (Guglielmi et al., 2020a). Furthermore, the Opalinus Clay features quite high attenuation of seismic waves, in particular in region where fractures and cracks exist (Nicollin et al., 2008): hence even for the tests where a clear geomachanical response is observed (FOP test – Fig. 11), the 2.5 m minimum distance between injection point and piezo-sensor could be already enough to damp the high frequency at which the acoustic emission should be observed.

## 6. Outlook for the long-term injection and implication for large scale storage

Based on the hydraulic observations, the pressure response suggests that the highest probability of eventual flow connection exists between injection interval Q4 and monitoring interval M1 and M2. Therefore, the long-term injection began in June 2019 (Phase 2) from the Q4 injection interval in the uppermost part of the fault with a mixture of $CO_2$-saturated water and a Kr tracer at a constant pressure of 4.5 MPa (below the FOP).

The long-term injection at CS-D will allow shedding light on several points that are listed below.

- The injection tests in Phase I did not result in strong changes in fault transmissivity, this does not necessarily hold for a long-term injection. Indeed, continuous pressurization could eventually weaken the fault, inducing seismic events or aseismic deformation that could enhance the permeability on the fault. Results from previous experiments (Guglielmi et al., 2020a), data from the $CO_2$ demonstration site at In Salah, Algeria (Rinaldi et al., 2017; Shi et al., 2013), as well as numerical modelling at medium to large scale (Rutqvist et al., 2016) clearly show that this could happen in caprocks. The CS-D system is designed to capture both seismic (acoustic emission) and aseismic deformation (e.g. at the SIMFIP). At the same time, it will be even more relevant to detect for healing mechanisms that would act against the leakage trough the fault. The Opalinus Clay is known to have strong swelling capabilities resulting in porosity decrease, and only a fully detailed monitoring of hydro-mechanical processes will allow to detect to what degree the fault zone strengthen in time.

- While a one-year experiment might not allow capturing all the geochemical processes involved in long-term injection operation, we expect a description of the processes at spatial and temporal resolutions that have not yet been explored. The geochemical monitoring system will allow for: i) continuous physico-chemical measurements at the monitoring boreholes (pH, electrical conductivity); ii.) fluid sampling that will be used to explore the evolution in major ion concentrations and isotopes (carbon, oxygen, hydrogen); iii.) determination of dissolved gas concentrations using an in situ portable mass spectrometer. This experimental setup will not only allow the monitoring the breakthrough of $CO_2$, but also the quantification of mixing with resident fluids and the description of the main geochemical interactions between the fluids and the rock.



- Given the decameter scale of the experiment, and its long duration in time, it is possible to test advanced scientific instrumentation in the long term. In particular two instruments that strongly enhance the monitoring capabilities at future large scale sites are: i) the SIMFIP, a probe capable of measuring three-dimensional deformation of the entire fault zone; ii) the "miniRuedi", a portable mass-spectrometer that allows for monitoring of the possible $CO_2$
breakthrough at given interval at depth. The highly dense network of instruments will allow to fully characterize the fluid dynamic at an unprecedented level of details for a fault zone in a caprock.  While the results of Phase I highlight a localized flow, it has to be seen in the long term of the $CO_2$ -saturated water distributes in the low permeable medium. Understanding the dynamic of the system will allow to better understand processes relevant for enabling safe underground $CO_2$ storage.

- Geophysical measurements will further strengthen the point above. Given the dense active seismic sensors network, regular survey will allow to monitor for changes in phase flow, in particular if the $CO_2$ start exsolving and saturate with gas the fault zone. Geophysical characterization has proved successful for the case when large features are present in caprock. An important example is the case of In Salah (Algeria), where a major fracture zone penetrating the caprock was highlighted by seismic surveys and analysis of deformation (Vasco et al., 2018).  In addition to
seismic measurement, we will perform regular rock electrical resistivity measurements to allow imaging any changes in fluid properties: already the difference between the in-situ water and $CO_2$-saturated synthetic water could create enough contrast in the decameter scale of the experiment. Regular monitoring at CS-D will allow for time-lapse images with detailed spatial and temporal resolution: in this way, we will be able to reach the lowest threshold to detect fluid flow in low permeable formations. If the active seismic analysis and the electrical
resistivity tomography will not be able to detect variation in the long-term injection, it could mean that the flow is confined in tiny fracture or that exsolution of the $CO_2$ is not strong, implying then that those additional monitoring techniques should be employed in combination at large scale storage sites. Finally, the CS-D experiment is a first-of-its-kind for duration and scale, and its permanent installation will allow for more than one single test. The CS-D experiment, and its successive series of tests, will produce a considerable amount of data that are essential for a proper
calibration of numerical models. With a data-driven approach, they will help to fill the gap between observed changes in rock permeability and modelling. Moreover, the use of Machine Learning (ML) as a predictive tool could play an essential role in enabling action to be taken to prevent failure of the sealing capacity of the storage/disposal site. Indeed, the long-term monitoring of injection, with regular hydro-mechanical characterization, will allow for time series with a considerable amount of data, enabling application of ML to prediction the time to failure.





7. **Conclusions**

In this paper, we describe the setup of the *in-situ* CS-D experiment at the MTRL. We drilled and instrumented a series of boreholes to perform long-term (one year) experiments and study sealing and induced seismicity related to leakage through low permeable faulted caprocks. The decameter scale experimental setup allows for close monitoring of fluid injected into a fault zone in the Opalinus Clay, simulating leakage through a faulted caprock at shallow depth. We installed geophysical, hydraulic, geomechanical and geochemical instrumentation that enable monitoring of several of the thermo-hydro-mechanical-chemical processes that occur at reservoir depth. In particular, our monitoring capabilities profited from two innovative instruments are: i) a probe capable of measuring three-dimensional deformation of the entire fault zone; ii) a portable mass-spectrometer that allows for monitoring of the possible $CO_2$ breakthrough at given interval at depth.

We also present here the results of the site characterization (Phase I) that highlight the complexity and uniqueness of the experiment.

Some of the aims of the experiment listed in chapter 2.1 could be already in the characterization phase. One of the aims of the CS-D experiment was to better understand the mobility of $CO_2$-rich water through the fault. Structural mapping and hydraulic characterization show the fluid does not flow preferentially along the fault but it is confined in small regions/fractures crosscutting the fault, even if the estimated permeability is extremely low.

Another target of the experiment was to test the occurrence of induced micro-seismicity in clay. No induced seismicity have been observed, even at injection pressure higher than the Fault Opening Pressure determined in previous experiments.

The seismic characterization successfully highlights the fault zone as a region of low velocity anomaly, probably due to different seismic velocity anisotropy in and outside the fault. Nevertheless the resolution does not allow to image small fractures through which the fluid flow may occur.

Many aims of the experiment (evolution of permeability after long-term exposure to $CO_2$, variation of geomechanical response with time) remain unresolved and only the analysis of data related to the second phase, the long-term injection of $CO_2$ saturated water would probably give some answer.

We think that the results from Phase I have already added some important observations to take into account for large scale $CO_2$ storage. The long-term injection of $CO_2$-saturated water will further elaborate the implication in terms of monitoring at large scale, and providing important parameters to validate numerical simulations, but also in terms of estimating the risk for $CO_2$ leakage at shallow depth.

**Code and Data availability**

Codes for numerical modelling are copyright of the LBNL. Code for analysis of injection tests are available at:
https://github.com/UniNE-CHYN/hytool. All data analysed in this paper are available under request from
alba.zappone@sed.ethz.ch.

**Sample availability**

Core samples could be obtained by submitting a proposal for experiment at claudio.madonna@erdw.ethz.ch.

**Acknowledgements**

The CS-D experiment is part of the ACT ELEGANCY, Project No 271498. ELEGANCY has received funding from DETEC
(CH), BMWi (DE), RVO (NL), Gassnova (NO), BEIS (UK), Gassco, Equinor and Total, and is co-funded by the European
Commission under the Horizon 2020 programme, ACT Grant Agreement No 691712. This project is supported by the Pilot
and Demonstration Programme of the Swiss Federal Office of Energy (SFOE).

CS-D is co-funded by the swisstopo, Chevron, Total, and receives in-kind contribution from the Lawrence Berkeley National
Laboratory.

We are thankful to swisstopo, and in particular to Paul Bossart, David Jaeggi, Senecio Schefer, Thierry Theurillat and the
Mont Terri rock laboratory team for onsite support. We thank Prof. Hans-Ruedi Maurer, from Earth Science Department in
ETH Zurich, for productive and cooperative discussions on data analysis. We are grateful to Prof. Rolf Kipfer, Dr. Matthias
Brennwald from EAWAG, and Ulrich W. Weber  from University of Uppsala, for providing support for dissolved gas
monitoring, and for data discussion. We thank Demir Semih Baris, Madalina Jaggi, Maria Kakurina, Nils Knornschild, Marija
Lukovic, Linus Villiger, Thomas Mörgeli, Michelle Robertson and Chet Hopp for technical support, scientific advice, and/or
on-site assistance.
Key contractors of the CS-D experiment are:
-   GeoSonic France, for the drilling of the boreholes
-   Solexperts Switzerland, for the design, installation, operation and maintenance of the injection equipment
-   Terratec Germany, for borehole logging.




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





| Borehole-ID | Main Purpose | Diameter [mm] | Length [m] | Inclination [°] | Azimuth [°] |
|---|---|---|---|---|---|
| **BCS-D1** | Injection | 101 | 23.4 | 0 | 0 |
| **BCS-D2** | Fluid sampling, Pressure, pH and EC monitoring | 101 | 18.6 | 0 | 0 |
| **BCS-D3** | Active source seismic and ERT tomography | 131 | 31.3 | 43 | 324 |
| **BCS-D4** | Active source seismic and ERT tomography | 131 | 36.4 | 42.5 | 323 |
| **BCS-D5** | Micro-seismicity monitoring and active seismic | 146 | 31.7 | 41 | 318 |
| **BCS-D6** | Micro-seismicity monitoring | 131 | 36.6 | 0 | 0 |
| **BCS-D7** | Slip monitoring | 116/101 | 30.5 | 0 | 0 |

**Table 1: Parameters defining the CS-D boreholes: Inclination= deviation from the vertical (vertical = 0°); borehole depth is approximate 0.5m. The order of the boreholes in table represents the drilling sequence. The borehole BCS-D7 has an initial diameter 935 of 116 mm from 0 to 13.60, and a diameter of 101 mm from 13.60 to 30.5 due to overcoring to retrieve a stuck drill bit.**





| | BCS-D1 | BCS-D2 | BCS-D4 | BCS-D5 | BCS-D6 | BCS-D7 |
|---|---|---|---|---|---|---|
| Top [m] | 14.34 | 11.04 | 27.05 | 19.74 | 28.50 | 22.46 |
| Dip dir [°] | 133.40 | 125.70 | 158.60 | 334.10 | 121.80 | 126.70 |
| Dip [°] | 65.80 | 55.20 | 64.00 | 81.10 | 59.50 | 64.20 |
| Bottom [m] | 19.63 | 16.39 | 28.44 | 22.66 | 31.40 | 25.54 |
| Dip dir [°] | 138.60 | 138.20 | 149.80 | 155.30 | 123.60 | 150.30 |
| Dip [°] | 62.00 | 56.70 | 63.50 | 66.00 | 59.40 | 55.90 |

**Table 2: Main Fault geometry: Top = top of the Main Fault (middle point), Bottom =Bottom of the Main Fault (middle point); dip**
**dir= direction from true North; due to poor image log quality in BCS-D3, no orientation could be identified.**

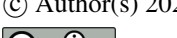



| Interval | Test | Date (UTC+1) | Pressure range (MPa) | Comments | Est. T (m²/s) |
|---|---|---|---|---|---|
| **Q1** | PST | 11.03 | 1.2-2.0 | Two series of ramp-up/ramp-down, then increased to 2 MPa and decay | N.E. |
| | SST | 17.04 | 1.25-4.8 | Steps 0.3 MPa every 10 minutes. | N.E. |
| | HP-SST | 17.04 | 4.8-6.0 | Steps 0.15 MPa every 5 minutes. | N.E. |
| | LST | 15:48 17.04 - 14:30 18.04 | 4.8 | Gradual step-down from HP-SST then single step for about 24 hours | N.E. |
| **Q2** | SST | 04.02 | 1.0-3.8 | Steps 0.2 MPa every 5 minutes. 20 minutes steps at 3.2 MPa and 3.8 MPa | N.E. |
| | LST1 | 18:57 27.02 – 00:35 10.03 | 1.2-3.6 | Steps 0.3 MPa every 28-30 hours. | N.E. |
| | LST2 | 15:56 28.03 – 12:40 16.04 | 1.8-4.8 | Steps 0.3 MPa every 28-30 hours. Last step lasted 172 hours. | N.E. |
| | HP-SST | 16.04 | 4.8-6.0 | Steps 0.15 MPa every 10 minutes. | N.E. |
| **Q4** | PST1 | 04.02 | 0.9-1.7 | Two series of ramp-up ramp-down, then increased to 1.7 MPa and decay | $1.8 \times 10^{-13}$ (decay) |
| | LST | 14:02 11.03 – 28.03 11.34 | 1.2-4.8 | Steps 0.3 MPa every 28-30 hours. | $2.8 \times 10^{-13}$ (first step) |
| | HP-SST | 28.03 | 4.8-6.0 | Steps 0.15 MPa every 10 minutes. Last step for 1.5 hours. | $4.0 \times 10^{-13}$ (decay) |
| | PST2 | 16.04 | 1.8-4.2 | Steps 0.3 MPa every 10 minutes. | $6.8 \times 10^{-13}$ (decay) |
| | PST3 | 11.06 | 1.2-4.8 | Steps 0.3 MPa every 10 minutes. **Fault opening pressure (FOP) reached**. | $9.2 \times 10^{-12}$ (decay) |

**Table 3: Summary of all injection tests performed. Transmissivity has been estimated only for tests in injection interval Q4 of borehole BCS-D1 (N.E. stands for not estimated). We calculated transmissivity only for this interval, as only when injection occurred here there was a response in the monitoring borehole (BCS-D2). Note that the transmissivity was always estimated by modelling the**
**injection pressure and never as cross-hole response**


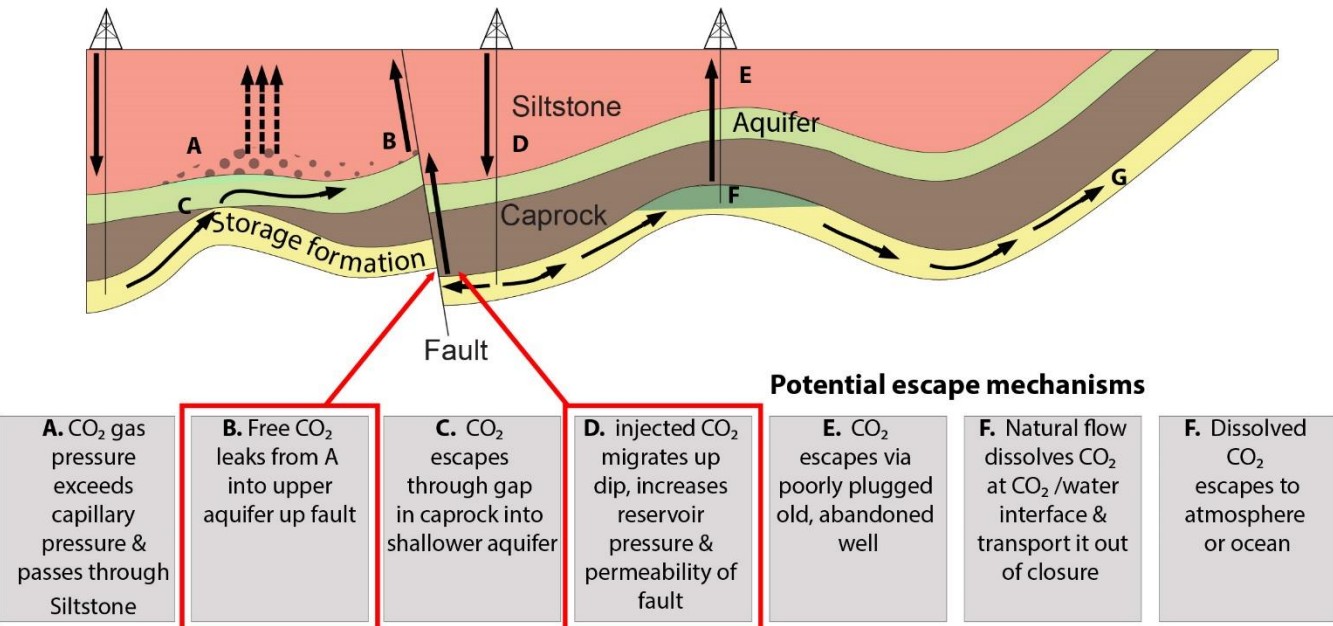

**Figure 1: Potential leakage routes (modified after IPCCP Special report 2005). The scientific objective of the experiment is to better understand how the prolonged exposure to CO₂-rich water could affects the properties of a fault hosted in a caprock, altering its sealing properties (red arrows).**


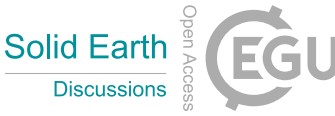

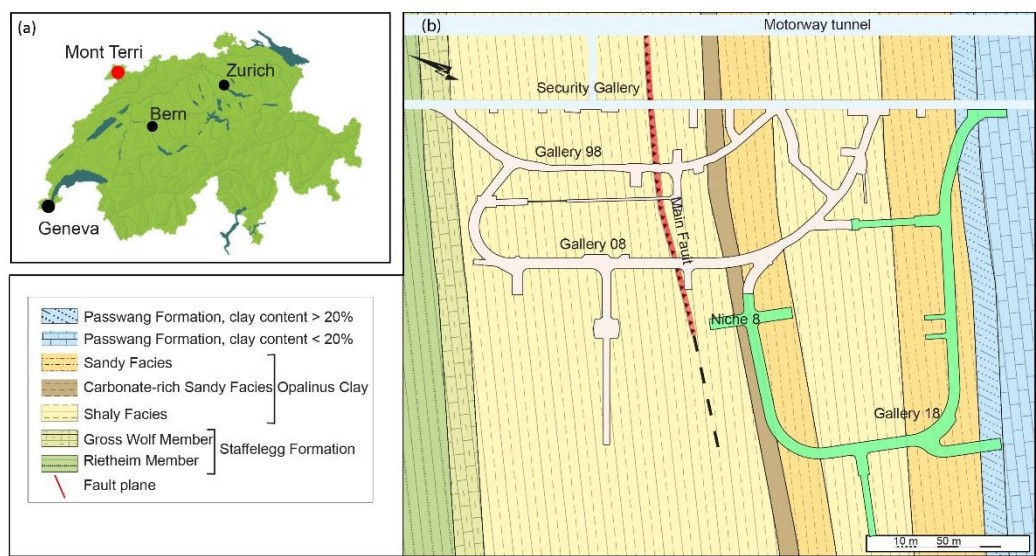

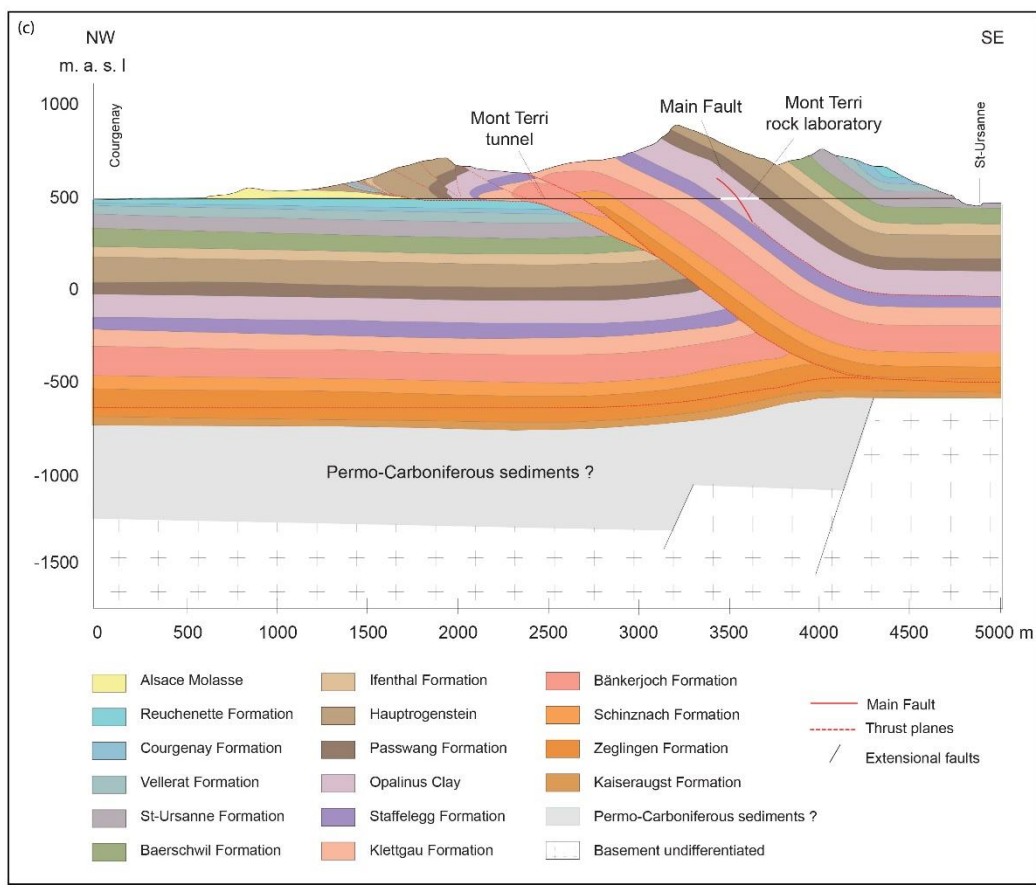



**Figure 2: a) Location of the MTR. b) Schematic geological map of the laboratory area with the new tunnels in light green (after Trury and Bossart 1999). c) Geological interpretation of the main structures along a profile crosscutting the security gallery**
**(Nussbaum et al., 2017).**

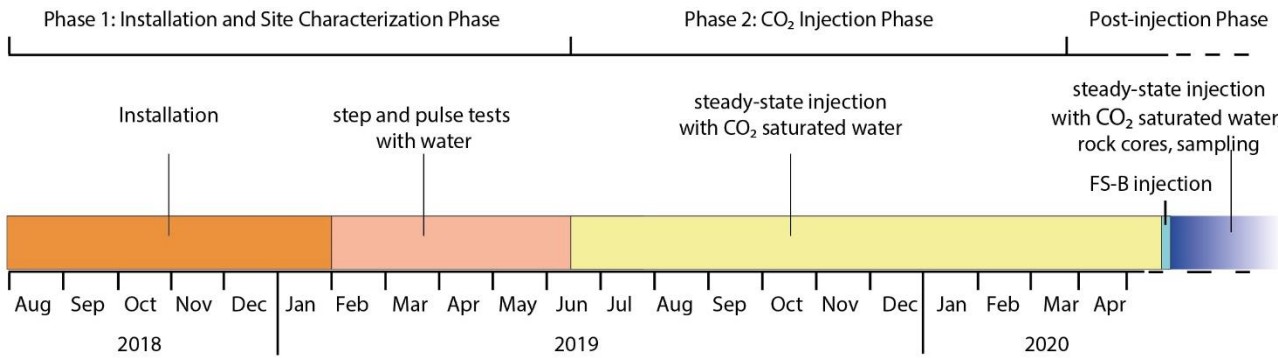

**Figure 3: timeline of CS-D (the FS-B injection is at date postponed to a later stage).**





Figure 4: (a) Modelling domain and main properties. (b) The pressure at injection follows a step-wise behaviour with monthly increase. (c) Example of simulated distribution of pressure changes around the injection point after 8 months of constant head injection for permeability of $5 \cdot 10^{-20}$ m². (d) Simulated injected brine distribution after 8 months of constant head injection for permeability of $5 \cdot 10^{-20}$ m².





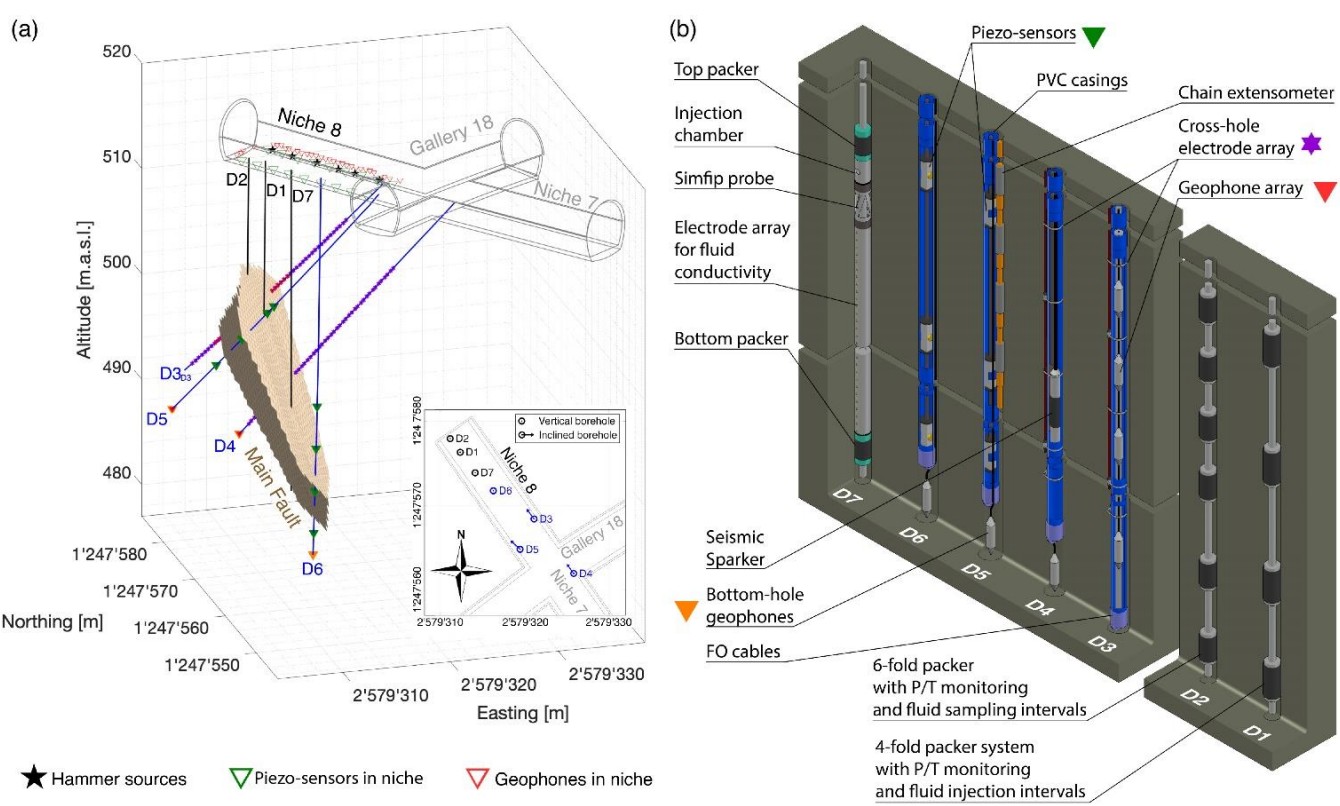


**Figure 5: a) Geometry of boreholes and of the Main Fault below niche 8;  b) planar view of the borehole location in niche 8.**





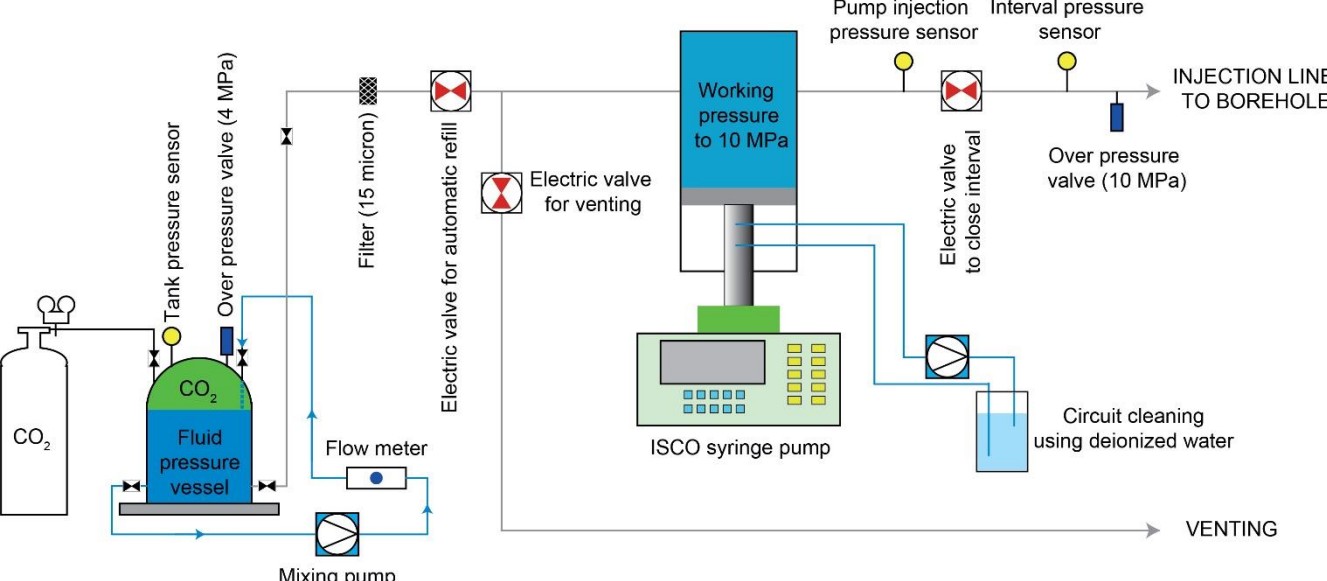

**Figure 6: Injection system design for the CS-D experiment (modified after Solexperts AG, Switzerland)**




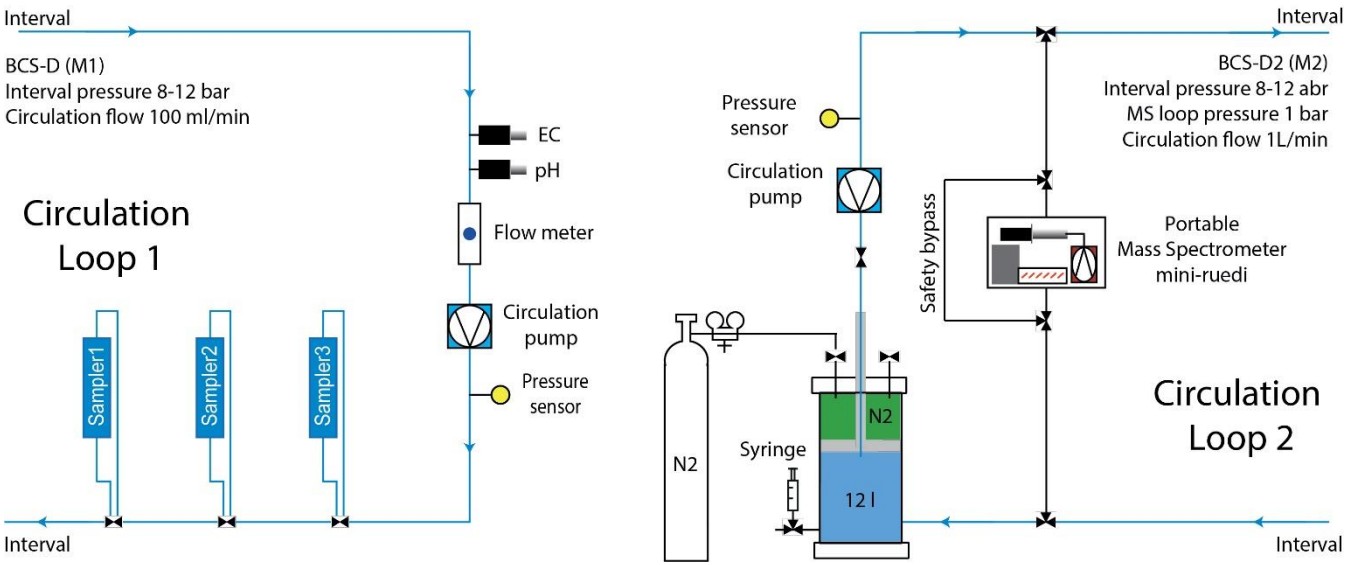

**Figure 7: Circulation loops for the CS-D experiment (modified after Solexperts AG, Switzerland).**






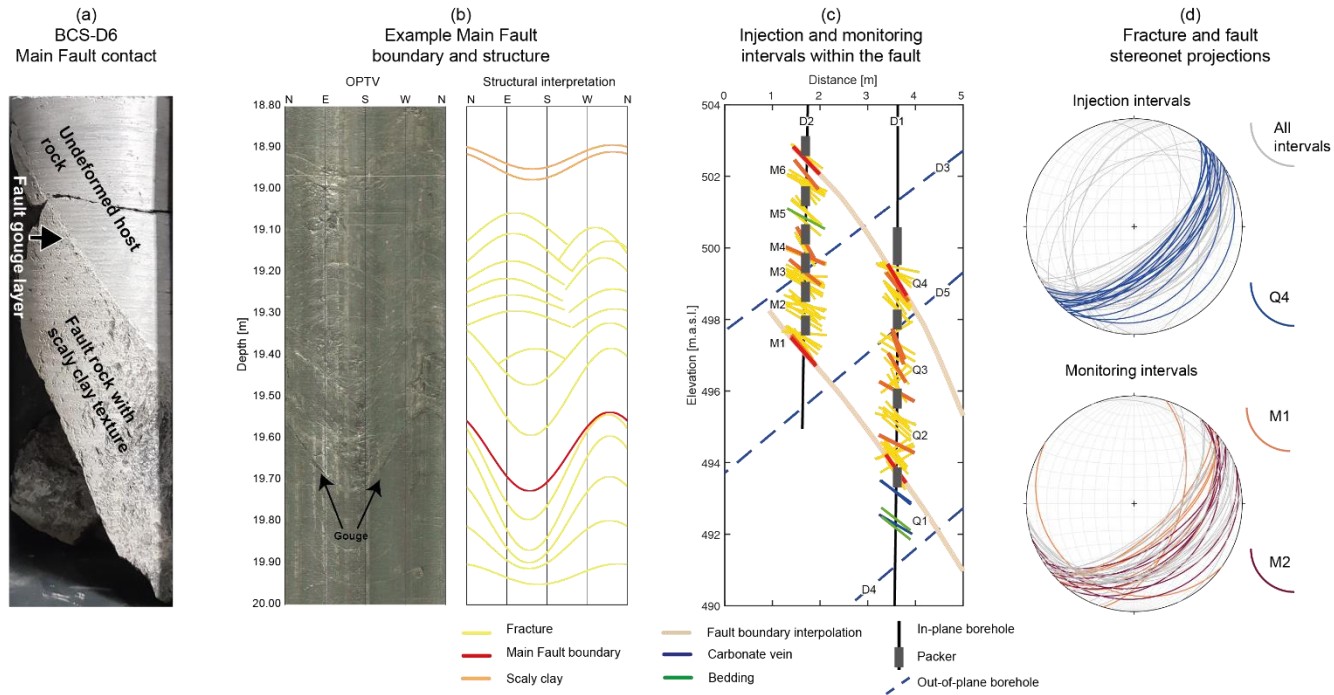

**Figure 8: a) An example image log of the contact between the Main Fault and the host rock is shown. b) An example of the contact between the host rock and the Main Fault from core. c) Injection and monitoring intervals and mapped structures within the Main Fault boundaries. d) Fracture and fault stereonet projections within the injection borehole (top) and monitoring borehole (bottom). The stereonets highlight all the structures within the Main Fault (grey) and the structures from the injection interval Q4 (blue), monitoring interval M1 (orange), and monitoring interval M2 (red).**


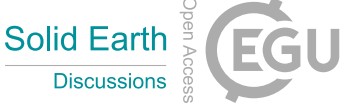



**Figure 9: a) Projection on plane of boreholes BCS-D1 and BCS-D2 and their intervals. The inlet shows the location of the boreholes and the orientation of the plane. b) Pressure changes in the injection interval Q4 (blue) and response at two intervals in the fluid monitoring borehole (M1 in red and M4 in green). The positions of the monitor intervals with respect the injection interval is shown in panel (a). c) Flow rate at the syringe pump.**







**Figure 10: a) Pressure decay after pulse in interval BCS-D1-Q4 (PST1). b) Analytical model and model derivative for pressure decay (Renard, 2017) and estimated transmissivity. c) First step of test LST in interval BCS-D1-Q4 (red is the flow rate, blue the pressure at injection). d) Analytical model of discharge (Renard, 2017) and estimate transmissivity.**






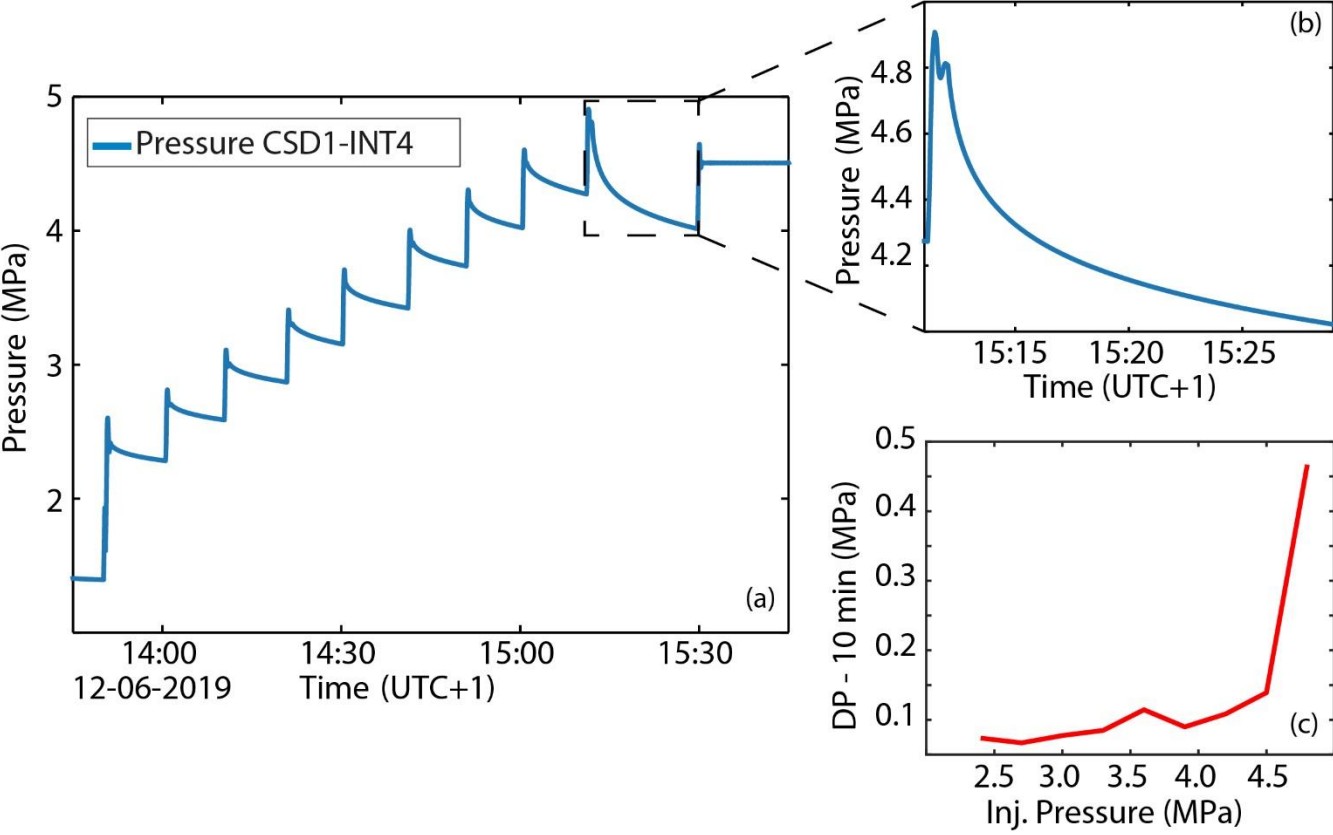

**Figure 11: a) PST in interval BCS-D1-Q4 to determine fault opening pressure (FOP); b) enlarge view of the pressure drop after reaching opening conditions; note that the decay curve is much larger when the FOP is reached; c) pressure changes after the 10 minutes step vs injection pressure: the system is non-linear above 4.5 MPa injection pressure.**







**Figure 12: (a) Position of the SIMFIP probe across the fault zone in borehole BCS-D7. (b-d) Long-term fault zone displacement and pore pressure monitoring: (b) Interval pressure, (c) Packer pressure, (c) (EW, NS, Z) displacement of the upper packer of the SIMFIP probe (Fault hanging wall). (E, f) Enlargement of relative and detrended displacement monitoring in stable periods before and after packer pressure calibration.**



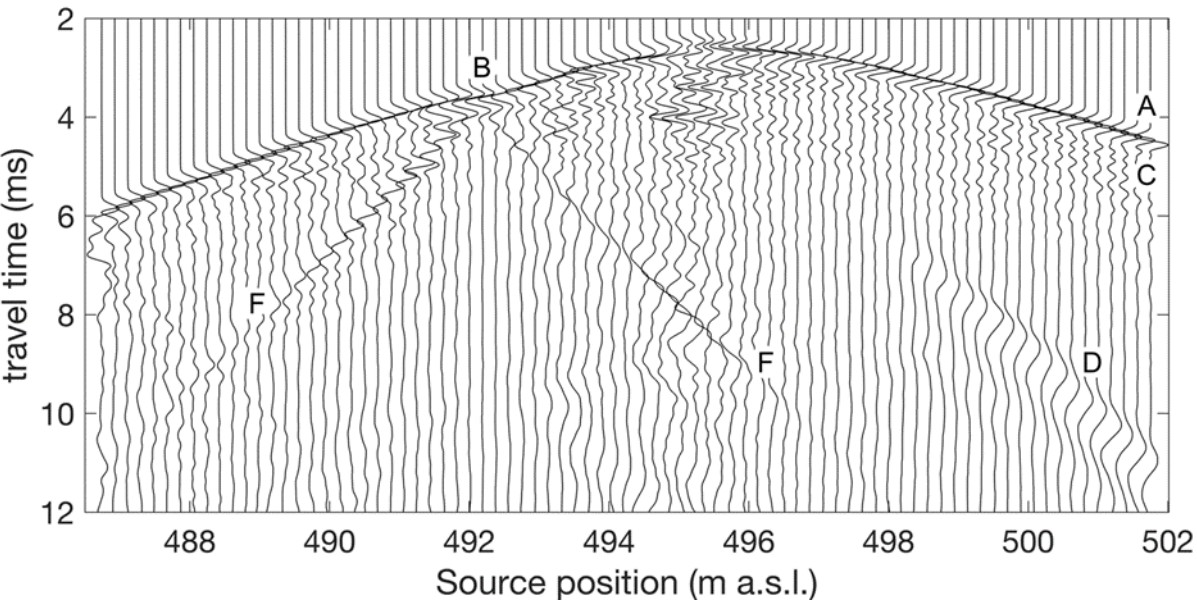

Figure 13: Common-receiver gather of processed seismic data, recorded with a cemented geophone in borehole BCS-D3, while a P-wave sparker source was employed at 25 cm intervals in borehole BCS-D4.





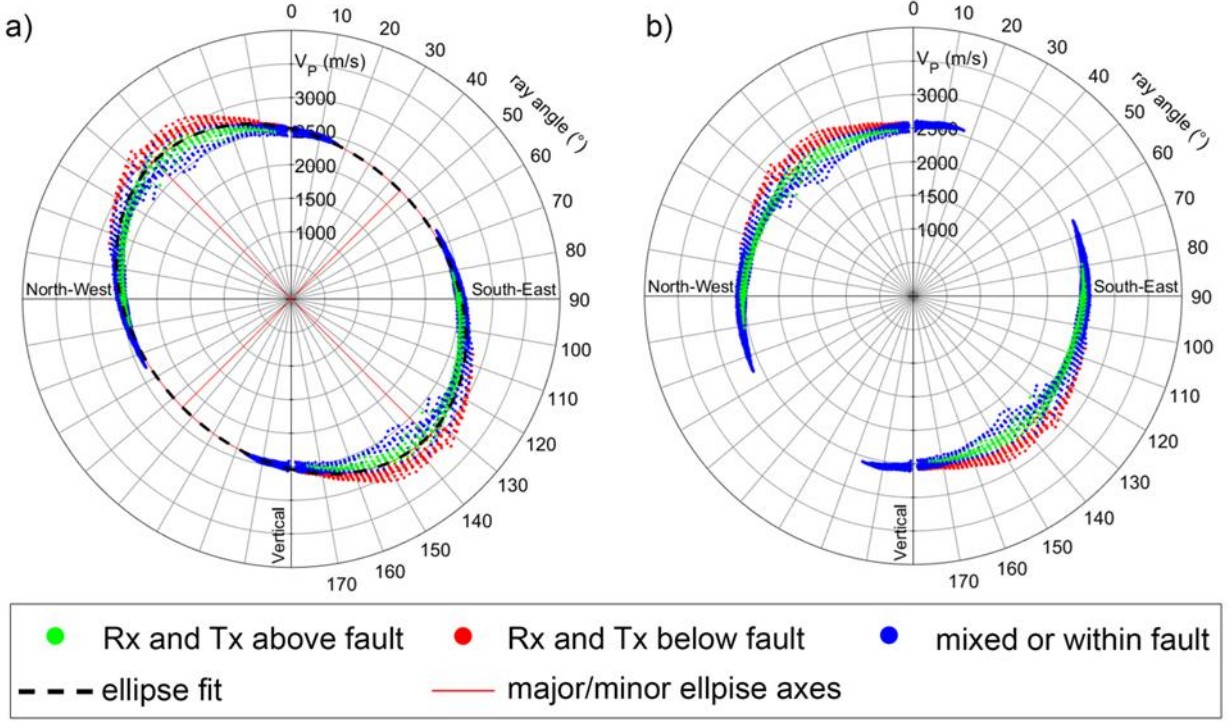

**Figure 14: Average V$_P$ for all receiver (Rx)-transmitter (Tx) pairs within the same plane as the one displayed in Figure 15. Since V$_P$ shows a strong anisotropy (a), velocities were normalized (b) prior to performing the travel time inversion.**





**Figure 15: V$_P$-tomogram obtained by cross-hole travel time inversion between borehole BCS-D3 (geophones) and BCS-D4 (sparker sources). Locations of geophones and sources are indicated by triangle- and star-symbols, respectively**
