# Peer review of "Fault sealing and caprock integrity for CO2 storage: an in-situ injection experiment"

_Solid Earth, 2020_

## Referee Comment (RC1) · Anonymous Referee #1 · 17 Sep 2020

General comments

In this paper, the experimental concept and initial results of a CO2 injection experiment at the scale of a rock laboratory are presented. It is the CS-D experiment within the Mont Terri Rock Laboratory which has a complex and comprehensive layout which are described in detail in the paper. Results are presented from the initial experimental phase, mainly focusing on the characterization of the fault system into which CO2 injection is carried out. The paper is overall well written with some minor technical issues I will list below.

The scientific background, including further related experiments and the current gap of knowledge (between lab scale and full size scale of a real storage site) are comprehensively described. Many technical details of the injection experiment and the monitoring

[Figure]

setup are given and presented with the support of well organized figures.

One striking issue is that in the discussion, the authors are claiming to provide conclusions which are valid for large scale storage. This aspect is, in my opinion, a bit over-estimated here. The main results of the experiment, to date, provide some very interesting insight into detailed process understanding related to leakage into fault zone, basically scale independant. The results especially of long-term injection into the fault will be highly interesting for the safety assessment of full scale storage sites, but these results are yet to come.

Also, a direct transfer of monitoring approaches from such small scale experiments to the full scale of large storage sites is problematic. The direct geophysical imaging of leaking $CO_2$ within the caprock will probably not be aimed at by operators due to the very high resolution needed at relatively large depth. The assessment of containment will rather be performed indirectly by potentially monitoring indicator horizons, such as aquifers right above the first caprock where leaking $CO_2$ may be accumulating. I would suggest to sharpen the statements in the general discussion a bit in this sense.

Specific comments:

1. Line 669 and ff.: "If the active seismic analysis and the electrical resistivity tomography will not be able to detect variation in the long-term injection, it could mean that the flow is confined in tiny fracture or that exsolution of the $CO_2$ is not strong, implying then that those additional monitoring techniques should be employed in combination at large scale storage sites."

Maybe I am missing something here but I cannot follow this logic. If seismic and ERT do not detect variation, they should be applied at large scale storage sites?

I guess you want to stress the recommendation to combine seismic and ERT because of their complementary sensitivity. That would enhance the chance to detect variations and to actually image a $CO_2$ related signature.

2. Line 700 and ff.: "The seismic characterization successfully highlights the fault zone as a region of low velocity anomaly, probably due to different seismic velocity anisotropy in and outside the fault."

Why do you argue with different anisotropy here. I think this is not necessary. It is reasonable to assume that the fault zone is mechanically weaker than the surrounding rock mass and thus seismic velocities are lower here. Anisotropy may be an additional factor, but not necessarily.

3. Seismic Tomography: Data processing: As the first step, you indicate that a median filter has been applied. What has been the purpose of this filter at this point? Removing secondary (linear) tube waves? Events before the P-wave first arrivals? Could this median filter affect the exact picking of traveltimes (time-shift as a processing artefact)?

Technical comments:

Line 17: confinement → containment Line 23: think → thick Line 99: delete one of the two ", ," Line 115: "pressure conditions exceeding the rupture of the fault" what does this mean? You mean pressure conditions causing fault rupture? Line 116: extent Line 125: by a synchronized... Line 169 f: Better write as a complete sentence: The CS-D experiment focuses on.... and addresses.... Line 172: ...it simulates... Line 177: north-wester (use identical spelling in the paper (northwester, north-wester)) Line 183: ...properties of a pristine claystone.... Line 187: that Based → of which, based ..., three main Line 430: "still performed in interval Q4" - this means "still ongoing"? maybe better indicate the actual period to which you are referring to, e.g, "still ongoing as of Sept. 2020" (just as an example). This paper will probably be read in the future when the experiment is not active anymore. Or do you mean "also"? Line 431: showed → shows Line 438: affected Line 449: ˆ(-12) Line 470: stabilized Line 539: if → of Line 660 and ff. There are some grammar issues. Figure 4d: colour scale ranging from 0 to 1, but brine distribution in %. Should the colour scale range from 0 to 100 % ? Figure 9: M1 and M5 in a, but M4 in b and in text. M5 should read M4? This is at least what

the reader may think while reading. Please clarify. Figure 13: Please explain shortly the meaning of A – F in the figure caption.

---

## Referee Comment (RC2) · Anonymous Referee #2 · 5 Oct 2020

General comments

This manuscript describes an experiment of induced CO2 migration along a fault at an original scale, between the laboratory and the field scale. Instrumented with various hydraulic, geophysical and geochemical tools, this experiment is an important step in the development of the CCS technology for its commercial deployment and public acceptance. It will provide an amazing set of data and allow a better understanding of faults systems, CO2 leak's risk, coupled hydro-chemo-mechanical processes and ways of monitoring CO2 leaks in the subsurface. My comments and remarques are hence more related to the manuscript itself, its organization and writing, than the experimental work, which is an impressive piece of work. Generally, the text could have been shortened and written in a more concise way. For example, some sections, like

the monitoring systems, are introduced several times (e.g. l.120 to l.128; l.146 to l.159) before being actually presented in details (l.330 to 378). An important part of the manuscript is also about the planned experiments (phases 2 and 3), again introduced at several locations before being detailed in the methodology section and discussed in section 5. The manuscript would have gained in clarity by i) focusing more on presenting and analyzing the main findings of the characterization phase 1, ii) then describing the instrumentation planned for the actual in-situ experiments, and what questions this experimental work will allow to clarify, iii) but without spending so much time discussing potential results and outcomes of these in-situ experiment and monitoring.

The second comment relates to the geochemical aspects of the work. It would help to add a few more details or explanations in some places. The first one is on the initial content of CO2 in the injected fluid: it is not clear in the text how and how much of CO2 is mixed: i.e. are pCO2, alkalinity (and thus pH) constant and constrained in the injected fluid? I would guess so, but this is not clear l.290 to l.294. A second point relates to monitoring the CO2 leak by a 'CO2 breakthrough' using the miniRuedi mass spectrometer. Again, all is fine with the methodology used, but it would help to add one sentence stating that this mass spectrometer will measure, both partial pressure of dissolved CO2 and CO2 gas (gas being produced by the fact that the pressure at that depth becomes smaller than the initial pCO2). Also, by talking about CO2 breakthrough we could take it as 'breakthrough of (pure, here gas) CO2, which I think is not the case and the authors aims at detecting the dissolved CO2 from the injected CO2-rich water.

Specific comments

l.68 to l.70: the studies described in the given references are not on fault reactivation per se, though they are still relevant to appear in this manuscript. I suggest replacing with a sentence similar to: 'The coupling between chemical and mechanical processes have been studied in the laboratory (Le Guen [. . .])."

l.84 to l.88: there is actually a second field scale experiment in Australia at the CO2

[Figure]

Otway Research Facility. This is an ongoing experiment, for which, similarly to what is described in this paper, the site characterization and the design of the monitoring have been completed so far and the actual 'shallow release of CO2' experiment will happen next year. There are only a few conference abstracts and proceedings publicly available as of now, but several papers are on preparation. Below are the references available and the links to the project description:

Feitz et al., 2018, The CO2CRC Otway shallow CO2 controlled release experiment: Preparation for Phase 2, Energy Procedia, Volume 154, Pages 145-150, doi: 10.1016/j.egypro.2018.11.024

Feitz et al., 2018, The CO2CRC Otway shallow CO2 controlled release experiment: Geological model and CO2 migration simulations, GHGT-14, 21st - 25th October 2018, Melbourne, Australia.

Tenthorey et al., 2019, The CO2CRC Otway Controlled CO2 Release Experiment in a Fault: Geomechanical Characterisation Pre-Injection, Conference Proceedings, Fifth International Conference on Fault and Top Seals, Volume 2019, p.1 – 5, doi:10.3997/2214-4609.201902321

https://www.ga.gov.au/about/projects/resources/geological-storage-co2
https://co2crc.com.au/co2research/srd3-3/

I would suggest also to add the following review papers

Migration and leakage of CO2 from deep Geological Storage sites by Bush and Kampman published in the AGU monograph Geological Carbon Storage: Subsurface seals and caprock integrity DOI: 10.1002/9781119118657.ch14

Fluid-rock interactions in Clay-rich seals: impact on transport and mechanical properties by Skurtveit et al. in that same monograph

l.219 to 226: the tense used in this section makes us confused about whether this phase is still under planning or has already been performed. Both present and future

tenses are used, however it is stated that this phase started in June 2019 for 12 months, which is thus something that should be now finished.

l.292: injection water: that would be better to mention here what is this injection water instead of later l.302 to l.304

l.293: It would be useful for the readers to briefly explain why Kr is used here

l.343: 'the circulation prevents chemical precipitation in the interval'. This is not clear to me at all why, it may prevent deposition of precipitated minerals but not necessarily their formation if the fluid is supersaturated with respect to these minerals

l.409 to l.411: do you mean that these labs have performed experiments on that topic in a general context, or do they have specifically studied the rocks of this study? In any case references should be added

Technical comments

The manuscript is generally well written but there are a few sentences throughout the text that are grammatically incorrect, or difficult to understand. e.g. l.116 to l.117; l.171 to l.173

There are also some words missing, errors in verb constructions, spelling mistakes or use of inappropriate words. I encourage the authors to carefully read and edit the text e.g. (this is not an exhaustive list)

l.96. and after: 'transport/migration' -> transport and migration

l.168: 'large volume injection' -> large volumes of injection or large injection volumes

l.169: 'role of CO2' -> role of CO2-rich water

l.170: 'behaviour of clay to several months of' -> behaviour of lays exposed to several months of

l.172: 'it stimulate' -> it stimulates

l.187: 'is sequence of shales, that Based on' -> is a sequence of shales that is based on

l.210: 'the fault suddenly show' -> the fault suddenly shows

l.227: 'we will repeat characterization' -> we will repeat the characterization'

l.230: 'petrophysical and geo-mechanical, and' -> petrophysical, geo-mechanical, and

l.233: sections instead of chapters

l.257: 'less permeable than as indicated' -> less permeable than the value found

l.259: 'the case of permeability 5.10ˆ-20 mˆ2' -> the case of a permeability of 5.10ˆ-20 mˆ2

l.260: 'up to 8m distance': please rephrase

l.270: 'in a way which' -> in a way that

l.272: 'are oriented normal to bedding planes' -> are (oriented) normal to the bedding planes l.303: 'depleted of' -> depleted from

l.353: 'fluid remobilization' -> remobilized fluid

l.406: 'drilling induced features' -> no drilling-induced features'

l.422: '(BCS-D1), which' -> (BCS-D1), and which

l.467: 'increased of' -> increased by

l.539 to l.541: please rephrase because the sense of this sentence is not clear

l.541: 'comprises gouge' -> comprises a gouge

l.554: 'if this observation were' please fix the singular/plural

l.556: 'the poroelasticity is negligible' -> the poroelasticity response is negligible'

---

## Author Comment (AC1) · 11 Nov 2020

We thank the Referee #1 for careful reading of the manuscript and constructive criticisms.

What he/she defines as minor technical issues have been carefully considered. We agree with the referee on the fact that the experiment is designed to provide insight into process understanding related to leakage into fault zone, basically scale independent. In the discussion, we describe the implications of the experiment for an upscaling to the full operational scale, partially making the effort to better contextualize the experiment. Following the referee's suggestion, we have partly rewritten the outlook section and modifies parts of the conclusions to keep the paper less speculative on the future

results of the long term injection.

Answer to specific comments

All specific comments have been addressed: 1. Line 669 and ff.: The transfer of monitoring approaches from (our) decameter scale experiment to the full scale of large storage sites is not intended to be a direct translation but it is an investigation on the strength of coupling different monitoring techniques. Furthermore, the sentence was partly wrongly written: what we mean is that when the flow is confined, geophysical measurements may not be able to detect leakage, and thus they must be used in combination with other instruments (like the ones that can monitor geochemical indicators). We agree that the sentence is not clear, and to avoid speculation we removed it from the outlook. 2. Line 700 and ff.: the fault zone is characterized by a fine network of fractures and slip planes (fig. 8a). The presence of fractures to some degree has an influence on the bulk seismic properties of the rock, but it strongly depends whether they are open (then they will decrease seismic velocities) or completely locked, as we rather expect. In the second case, this effect is probably minor. The internal structure of the fault defines a different anisotropy pattern inside the fault juxtaposed to the host rock, thus strongly contributing to the seismic anomaly. The sentence has been reformulated. 3. Seismic Tomography: data processing: The median filter applied is a simple subtraction of the median amplitude (median of the amplitudes along the entire trace lengths) from the signal. Therefore, it has no other effect than removing a constant shift from the signal. The sentence has been modified.

All technical comments have been accepted and all amendments have been inserted in text and figures.

We hope that with the modifications we made the text can be accepted for publication.

Alba Zappone on behalf of the Authors
* * *

---

## Author Comment (AC2) · 11 Nov 2020

Answer to general comments

We thank the Referee #2 for careful reading of the manuscript and constructive criticisms. The comments and remarks of Referee #2 are related to the organization and writing of the manuscript, that could be written in a more concise way. We addressed this criticism and reorganized some sections. To give more emphasis on the phase 1 of the experiment, the parts related to the outlook and successive stages have been shortened. Nevertheless, we think that a description of the potential outcomes of the next steps and the implications of the experiment for an upscaling to the full operational scale, is beneficiary to better contextualize the experiment. We have partly rewritten

the outlook chapter and shortened the speculations projected towards the phases of the experiment following those described in the manuscript. Regarding the geochemical aspects presented in the work, we tried to add as much as possible details where it was suggested by the Referee #2. The mixing of the injection fluid and how its composition is determined is now better described. It is true that our installation has technical limits and we cannot define a priori the content of $CO_2$ in the injected fluid, but we calculate it through known parameters (pressure, temperature of mixing, and chemical composition of the fluid prior $CO_2$ saturation). A second point relates to monitoring the $CO_2$ leak using the miniRuedi mass spectrometer. We added a sentence detailing what is actually measured by the spectrometer and what we mean about $CO_2$ breakthrough, which is, as the referee correctly pointed out, the dissolved $CO_2$ from the injected $CO_2$-rich water.

Answers to specific comments

L. 68 to l.7 0: The text has been modified as suggested.

l.84 to l.88: We thank the Referee#2 for pointing out this omission of a second field scale experiment in Australia at the $CO_2$ Otway Research Facility, and for the indication of the relevant literature about it. The text of the manuscript has been completed, including this experiment, and of course we are thrilled to follow the development of this field scale experiment in Otway! The review papers by Bush and Kampman, 2018 and Skurtveit et al., 2018 have been added to the references.

l.219 to 226: the tense used in this section has been reviewed. The phase 2 started in June 2019 and is still ongoing. We think we managed to modify the tense in a more correct way.

l.292 : injection water: the composition of the water has been introduced here and not later l.302 to l.304: l.293 : we briefly explained that we used Kr as tracer because 1. It is non-reactive, 2. It is absent in formation water.

[Figure]

l.343: the circulation guarantees that the fluid is homogeneous in the sampling interval; the text has been amended.

l.409 to l.411 : ETH and EPFL labs have performed experiments specifically on the rocks of this study. References have been added.

Technical comments

All technical comments have been accepted and amendments have been inserted in text and figures.

We hope that with the modifications we made the text can be accepted for publication.

Alba Zappone on behalf of the Authors

─────────────────────────────